# Algorithms that Approximate Data Removal: New Results and Limitations

**Vinith M. Suriyakumar**
MIT EECS
vinithms@mit.edu

**Ashia C. Wilson**
MIT EECS
ashia07@mit.edu

## Abstract

We study the problem of deleting user data from machine learning models trained using empirical risk minimization (ERM). Our focus is on learning algorithms which return the empirical risk minimizer and approximate unlearning algorithms that comply with deletion requests that come in an online manner. Leveraging the infintesimal jacknife, we develop an online unlearning algorithm that is both computationally and memory efficient. Unlike prior memory efficient unlearning algorithms, we target ERM trained models that minimize objectives with non-smooth regularizers, such as the commonly used $\ell_1$, elastic net, or nuclear norm penalties. We also provide generalization, deletion capacity, and unlearning guarantees that are consistent with state of the art methods. Across a variety of benchmark datasets, our algorithm empirically improves upon the runtime of prior methods while maintaining the same memory requirements and test accuracy. Finally, we open a new direction of inquiry by proving that all approximate unlearning algorithms introduced so far fail to unlearn in problem settings where common hyperparameter tuning methods, such as cross-validation, have been used to select models.

## 1 Introduction

The right to be forgotten (RtbF) is considered a fundamental human right in many legal systems [27, 21, 5]. The proliferation of techniques that employ user data to do things such as training and validating machine learning across a variety of organizations has led to reconsideration of how to interpret RtbF. The European Union's General Data Protection Regulation (GDPR), California's Consumer Privacy Act (CCPA), and Canada's proposed Consumer Privacy Protection Act (CPPA) are all examples of new pieces of legislation which attempt to codify the RtbF by requiring companies and organizations to delete a user's data by request [34].

But *what does it mean to delete a user's data?* User data, for example, can be recovered from trained machine learning models [31] and hyperparameter tuning procedures [25]. This suggests that data deletion should require organizations to take action on the models and algorithms derived from the data as well. This interpretation is consistent with the Federal Trade Commission's recent action ordering companies to delete the data from users who deleted their account as well as the models and algorithms derived from the users' data [7]. Forcing organizations to comply with each deletion request by retraining, however, comes with potentially significant monetary and time costs. Thus, it is worth asking if these costs can be managed while still maintaining model performance.

The cost concerns associated with RtbF compliance have prompted several recent works which formalize and study the problem of unlearning [6]. The aim of these works is to develop techniques which delete user data from models that are inexpensive in both computation and memory while maintaining reasonable performance. Most unlearning algorithms proposed so far focus on unlearning models obtained via empirical risk minimization [14, 28, 4, 23, 20]. However, several of these methods are memory intensive [4, 15], requiring organizations to store several states of the model

36th Conference on Neural Information Processing Systems (NeurIPS 2022).

while training. We build on recent works [14, 28] which show that for models derived from minimizing convex and sufficiently smooth ERM objectives, a Newton update to the current model can be applied in order to approximately unlearn in certain data directions. We show a different Newton estimate based on the classical infinitesimal jacknife developed in the robust statistics literature [17, 10] results in a more computationally efficient unlearning algorithm in the online setting. While the current legislation only requires companies to satisfy delete requests in 30 days there are scenarios where immediate (online) delete requests are necessary such as the UK Biobank [12]. We also show how to utilize the proximal infinitesimal jackknife [35] to unlearn in settings when the model was trained using an ERM objective that is not smooth. Finally, we present an important and concerning failure mode of all approximate unlearning algorithms.

Our principal contributions are three-fold.

- *Computationally efficient online algorithm.* We develop an online unlearning algorithm to unlearn a sequence of $m$ datapoints from a model $\hat{\theta}_n(\lambda)$ that approximately minimizes a smooth and convex ERM objective that requires $O(d^2)$ storage and has running time $O(md^2)$. To unlearn $m$ datapoints in an online manner, the algorithms developed in prior works (e.g. [14, 28]) requires a similar storage, but have a longer $O(md^3)$ running time. By avoiding the cost of computing and inverting a different Hessian at each deletion request we improve on the computational requirements of previous memory efficient unlearning algorithms, while maintaining the same generalization, unlearning and deletion capacity guarantees. We empirically demonstrate significant computational savings without sacrificing test set accuracy on multiple datasets.

- *Accommodating non-smooth regularizers.* We provide a generalization of our unlearning algorithm based on the proximal Newton method that can be used to efficiently delete data from models that minimize objective functions with non-smooth penalties. We provide state-of-the art unlearning, deletion capacity and generalization guarantees for our new unlearning method. We empirically demonstrate significant computational savings without sacrificing test set accuracy in predicting Warfarin dosages.

- *Failure modes.* Most data processing pipelines are not as simple as writing down an objective function and running an algorithm out-of-the-box which returns the approximate empirical risk minimizer. A more common practice is to tune hyperparameters using techniques such as cross-validation. We reveal a fundamental limitation of all (approximate) data removal processes developed so far when hyperparameter tuning takes place.

## 2 Related Work

Given the increasing concerns around privacy of user data, recent research on *machine unlearning* studies how we can efficiently delete datapoints used to train models without retraining from scratch. This work was first initiated by Cao and Yang [6] whose definition requires the outputs of an unlearning algorithm to be identical to the outputs of the model produced by retraining. Since then, several works have provided different definitions of machine unlearning which can be separated as exact unlearning ([4, 32, 15, 11]) or approximate unlearning ([14, 28, 12, 9, 23]).

Our work is focused on satisfying approximate unlearning definitions inspired by differential privacy [9] defined by [14] as $(\epsilon, \delta)$-certified removal or by [28] as $(\epsilon, \delta)$-unlearning. This definition requires the output distribution of the unlearning algorithm to be similar to that obtained by retraining from scratch using the original training set with the requested datapoints removed from it. We focus on this definition because algorithms for exact unlearning such as statistical query learning [6], SISA [4], and its adaptive variant [15] have high computational and memory cost.

Prior algorithms developed for approximate unlearning use a variety of techniques: perturbed gradient descent [23, 33], Newton style updates [14, 28], and projected residual updates [16]. The perturbed gradient descent and projected residual update methods provide theoretical error guarantees for the empirical training loss but fail to provide any generalization guarantees. In addition, while these methods reduce the computation burden associated with exact unlearning, their high memory costs are still similar to exact unlearning methods.

Given these issues, Newton update unlearning algorithms were proposed to efficiently satisfy approximate unlearning. Our work is closest to [14, 28] who use Newton updates to efficiently delete data

for generalized linear models. Sekhari et al. [28] improve upon Guo et al. [14] in multiple ways: (i) by not requiring full access to the training dataset, providing (ii) generalization guarantees, and (iii) removing the requirement for randomization in the learning algorithm itself (which often reduces utility). Yet the algorithm developed by [28] targets the batch setting which is unrealistic in practice and they provide no empirical results demonstrating the efficacy of their algorithm. If implemented in an online way, their algorithm would suffer a much larger computational cost given the requirement to compute and invert a new Hessian for each delete request. Finally, neither of these algorithms provide theoretical guarantees for the commonly used models that are obtained from objective functions with non-smooth penalties.

We address several of these issues using the (proximal) infinitesimal jacknife [13, 17, 10]. Our algorithm can handle online delete requests selected in an adaptively adversarial manner (a failure mode of many previous algorithms pointed out by Gupta et al. [15]) making it more practical for real world use. Furthermore, we provide generalization and deletion capacity guarantees similar to Sekhari et al. [28] for both smooth and non-smooth regularizers. To do so, we leverage similar guarantees found in approximate cross-validation literature [35]. Finally, inspired by recent working showing that hyperparameter tuning can leak user data [25] we demonstrate that all approximate unlearning algorithms introduced so far fail to unlearn in settings where hyperparameter tuning has taken place to choose a model.

## 3 Methods and Results

**Learning**   Consider the objective function comprised of a loss function $\ell$, a regularizer $\pi$ and regularization parameter $\lambda \in \Lambda \subseteq [0, \infty]$. The goal of learning is to find a parameter $\theta^*(\lambda)$ which minimizes the population risk

$$F(z, \theta, \lambda) \triangleq \mathbb{E}_{z \sim \mathcal{D}}[\ell(z, \theta)] + \lambda \pi(\theta)$$

Given the distribution $\mathcal{D}$ is often inaccessible, practitioners often instead find a model $\hat{\theta}_n(\lambda)$ which (at least approximately) minimizes the empirical risk

$$F_n(z, \theta, \lambda) \triangleq \frac{1}{n} \sum_{i=1}^{n} \ell(z_i, \theta) + \lambda \pi(\theta,) \tag{1}$$

corresponding to a given dataset $S = (z_1, \ldots z_n)$.

**Unlearning**   Having used a dataset $S$ to train and publish a model $\hat{\theta}_n(\lambda)$, a set of $m$ users in the training set $U \subset S$ might request that their datapoints be deleted and that any models produced using their data be removed. To comply with this request, an organization might find the minimizer $\theta_{n,-U}(\lambda)$ of the leave-$U$-out objective

$$F_{n,-U}(z, \theta, \lambda) \triangleq \frac{1}{n-m} \sum_{z \in S \backslash U} \ell(z, \theta) + \lambda \pi(\theta). \tag{2}$$

While *reoptiminzing the leave-one-out objective from scratch constitutes a baseline for the problem of unlearning*, the computational cost makes complying with every data delete request in this way undesirable. Training from scratch satisfies the notion of unlearning [28] formalized in Definition 1.

**Definition 1** (($\epsilon, \delta$)-unlearning [28])**.** *Let $S$ be a fixed training set and $A : S \to \theta$ be an algorithm that trains on $S$ and outputs a model $\theta \in \Theta$. For an $\epsilon > 0$ and set of delete requests $U \subseteq S$, we say that a removal mechanism $M$ is ($\epsilon, \delta$)-unlearning for learning algorithm $A$ if $\forall \mathcal{T} \subseteq \Theta$ and $S \subseteq \mathcal{Z}$, the following two conditions are satisfied:*

$$P(M(U, A(S), T(S)) \in \mathcal{T}) \leq e^\epsilon P(M(\emptyset, A(S \backslash U), T(S \backslash U)) \in \mathcal{T}) + \delta, \quad and$$
$$P(M(\emptyset, A(S \backslash U), T(S \backslash U)) \in \mathcal{T}) \leq e^\epsilon P(M(U, A(S), T(S)) \in \mathcal{T}) + \delta$$

Finally, we point out that like most previously proposed unlearning algorithms, we do not require $T(S)$ to contain the entire training set, but instead propose unlearning algorithms for which $T(S)$ is independent of $n$.

## 3.1 Unlearning models obtained via regularized empirical risk minimization

We recommend use of the following proximal operator to comply with delete requests

$$\text{prox}_{\lambda\pi}^H(v) \triangleq \underset{\theta \in \mathbb{R}^d}{\arg\min} \|v - \theta\|_H^2 + \lambda\pi(\theta), \tag{3}$$

which allows us to handle objective functions that incorporate non-smooth regularizers, such as the $\ell_1$, elastic net or nuclear norm penalty. More specifically, having deleted the datapoints in the set $U$, we propose Algorithm 1 to delete the data of an additional user $j$.

---

**Algorithm 1** Infinitesimal Jacknife (**IJ**) Online Unlearning Algorithm

---

**Input:** $|U| = m$, Hessian of loss or objective, $\tilde{\theta}_{n,-U}(\lambda)$, $\bar{\theta}_{n,-U}^N(\lambda)$ $\hat{\theta}_n(\lambda)$, and delete request $j$

**Set:** $H_\ell = \frac{1}{n}\sum_{i=1}^n \nabla_\theta^2 \ell(z_i, \hat{\theta}_n)$, $H_f = \frac{1}{n}\sum_{i=1}^n \nabla_\theta^2 f(z_i, \hat{\theta}_n, \lambda)$, $\bar{\theta}_{n,-\emptyset}^N(\lambda) = \tilde{\theta}_{n,-\emptyset}(\lambda) = \hat{\theta}_n(\lambda)$

**If $\pi$ is not smooth:**

    **Compute:**

$$\bar{\theta}_{n,-(U\cup\{j\})}(\lambda) = \text{prox}_{\lambda\pi}^{H_\ell}\left(\bar{\theta}_{n,-U}^N(\lambda) + \tfrac{1}{n}H_\ell^{-1}\nabla\ell(z_j, \hat{\theta}_n(\lambda))\right) \tag{4a}$$

$$c = \frac{(m+1)^2(2CL\mu + ML)}{\mu^2 n^2}\frac{\sqrt{2\ln(1.25/\delta)}}{\epsilon} \tag{4b}$$

    **Store:** $\bar{\theta}_{n,-(U\cup\{j\})}^N(\lambda) = \bar{\theta}_{n,-U}^N(\lambda) + \frac{1}{n}H_\ell^{-1}\nabla\ell(z_j, \hat{\theta}_n(\lambda))$

**Else compute:**

$$\bar{\theta}_{n,-(U\cup\{j\})}(\lambda) = \tilde{\theta}_{n,-U}(\lambda) + \tfrac{1}{n}H_f^{-1}\nabla\ell(z_j, \hat{\theta}_n(\lambda)) \tag{5a}$$

$$c = \frac{(2m+1)(2CL\mu + ML)}{\mu^2 n^2}\frac{\sqrt{2\ln(1.25/\delta)}}{\epsilon} \tag{5b}$$

**Sample:** $\sigma \sim \mathcal{N}(0, cI)$

**Return:** $\tilde{\theta}_{n,-(U\cup\{j\})}(\lambda) := \bar{\theta}_{n,-(U\cup\{j\})}(\lambda) + \sigma$

---

**Guarantees** While we give guarantees for functions that are strongly convex, we rely on standard reductions from the convex setting to the strongly convex setting (i.e. based on defining a objective function $F(\theta) = \tilde{F}(\theta) + (\mu/2)\|\theta\|^2$ when $\tilde{F}$ is convex). Furthermore, similar to Guo et al. [14, 3.3] our methods can be used for unlearning in the non-convex setting when the deep learning model applies a simple convex model to a differentially private feature extractor [1].

**Assumption 1** (Smooth regularizer). *For any $z \in \mathcal{Z}$, the objective function $F(\theta, z, \lambda)$ is $\mu$-strongly convex and $L$-Lipschitz with $M$-smooth Hessian. The loss has $C$-Lipschitz Hessians.*

**Assumption 2** (Non-smooth regularizer). *For any $z \in \mathcal{Z}$, the loss function $\ell(\theta, z)$ is $\mu$-strongly convex and $L$-Lipschitz with $M$-smooth and $C$-Lipschitz Hessians. The regularizer $\pi(\theta)$ is convex.*

With either of these assumptions, it's possible to show that the denoised output of Algorithm 1 is $O(m^2/n^2)$ close to the exact unlearned model. We formalize this proximity result in the following Lemma 1.

**Lemma 1** (Proximity to the baseline estimator). *Suppose $F$ satisfies Assumption 1 or Assumption 2. Let $S$ be the dataset of size $n$ sampled from $\mathcal{D}$ and $U \subseteq S$ denote the set of $m$ delete requests. Consider $\bar{\theta}_{n,-U}(\lambda)$, i.e. the output of Algorithm 1 without noise term $\sigma$ added and the model $\hat{\theta}_{n,-U}(\lambda)$ obtained by minimizing the leave-$U$-out objective $F_{n,-U}$. Then,*

$$\|\hat{\theta}_n(\lambda) - \hat{\theta}_{n,-U}(\lambda)\|_2 \le \frac{mL}{\mu n}, \qquad \text{and} \tag{6a}$$

$$\|\hat{\theta}_{n,-U}(\lambda) - \bar{\theta}_{n,-U}(\lambda)\|_2 \le \frac{2m^2 CL}{n^2\mu^2} + \frac{m^2 ML^2}{n^2\mu^3}. \tag{6b}$$

Lemma 1 implies adding the noise term $\sigma \propto O(m^2/n^2)$ will result in the desired unlearning guarantee. The following Theorem 1 outlines this guarantee as well as the proximity of the unlearnt model to the test loss minimizer. The proof is contained in Appendix A.1.

**Theorem 1** (Unlearning and generalization guarantees). *Suppose the loss function satisfies Assumption 1 or Assumption 2 and consider any learning algorithm that returns a model $O(1/n^2)$ close to the empirical risk minimizer $\hat{\theta}_n(\lambda)$ trained on any dataset $S \sim \mathcal{D}$ of size $n$. Then the output $\tilde{\theta}_{n,-U}(\lambda)$ of Algorithm 1, where $|U| = m$, satisfies the test error bound*

$$\mathbb{E}[F(\tilde{\theta}_{n,-U}(\lambda)) - F(\theta^*(\lambda))] \leq O\left(\frac{(2m^2CL^2\mu + ML^3m^2)}{\mu^3 n^2} \cdot \frac{\sqrt{d}\sqrt{ln(1/\delta)}}{\epsilon} + \frac{4mL^2}{\mu n}\right).$$

*Furthermore, the unlearning Algorithm 1 results in $(\epsilon, \delta)$-unlearning of $\forall \mathbf{z} \in U \subseteq S$.*

### 3.1.1 Deletion Capacity

Sekhari et al. [28] introduce the notion of deletion capacity which formalizes how many samples can be deleted from a the model parameterized by the original empirical risk minimizer $\hat{\theta}_n(\lambda)$ while maintaining reasonable guarantees on the test loss. We restate the definition here.

**Definition 2** (Deletion capacity [28]). *Let $\epsilon, \delta, \gamma > 0$ and $S$ be a dataset of size $n$ drawn i.i.d From $\mathcal{D}$, and let $F(\theta, z, \lambda)$ be an objective function. For a learning algorithm and removal mechanism $M$ that satisfies $(\epsilon, \delta)$-unlearning of all $\mathbf{z} \in U$, where $U$ is the set of delete requests, the deletion capacity $m_{\epsilon,\delta,\gamma}^{A,M}(d,n)$ is defined as the maximum number of samples that can be unlearned while still ensuring the excess population risk is $\gamma$. Specifically,*

$$m_{\epsilon,\delta,\gamma}^{A,M}(d,n) \triangleq \max\left\{m \,|\, \mathbb{E}\left[\max_{U \subseteq S:|U| \leq m} F(M(A(S), S, U) - F(\theta(\lambda)^*)]\right] \leq \gamma\right\}$$

*where the expectation is with respect to $S \sim \mathcal{D}$ and output of the learning algorithm $A$ and removal mechanism $M$.*

Sekhari et al. [28] provide both an upper bound and lower bound on the deletion capacity of unlearning algorithms. We recount both bounds and show our algorithm achieves the same bounds.

**Theorem 2** (Deletion capacity upper bound [28]). *Let $\delta \leq 0.005$ and $\epsilon = 1$. There exists a 4-Lipschitz and 1-strongly convex loss function f, and a distribution $\mathcal{D}$ such that for any learning algorithm $A$ and removal mechanism $M$ that satisfies $(\epsilon, \delta)$-unlearning and has access to all undeleted samples $S \backslash U$, then the deletion capacity is:*

$$m_{\epsilon,\delta,\gamma}^{A,M}(d,n) \leq cn,$$

*where the constant $c$ depends on the Lipschitz constants $L$, $M$, strongly convex constant $\mu$, and boundedness constant $C$ from Assumptions 1 or 2 and $c < 1$.*

**Theorem 3** (Deletion capacity lower bound [28]). *Let $\epsilon, \delta > 0$ and $\gamma = 0.01$, $S$ be a dataset of size $n$ drawn i.i.d from $\mathcal{D}$, and $F(\theta, z, \lambda)$ be an objective function satisfying Assumption 1 or Assumptions 2. Consider a learning algorithm $A$ that returns the empirical risk minimizer and unlearning algorithm $M$. Then the deletion capacity is:*

$$m_{\epsilon,\delta,0.01}^{A,M}(d,n) \geq c \cdot \frac{n\sqrt{\epsilon}}{(d\log(1/\delta))^{\frac{1}{4}}} \tag{7}$$

*where the constant $c$ depends on the Lipschitz constants $L$, $C$, and $M$.*

**Theorem 4** (Deletion capacity from unlearning via DP [28]). *There exists a polynomial time learning algorithm $A$ and removal mechanism $M$ of the form $M(U, A(S), T(S)) = A(S)$ such that the deletion capacity is:*

$$m_{\epsilon,\delta,0.01}^{A,M}(d,n) \geq \tilde{\Omega}\left(\frac{n\epsilon}{\sqrt{d\log(e^\epsilon/\delta)}}\right)$$

*where the constant in $\Omega$-notation depends on the properties of the loss function $f$.*

As an extensions of a result developed by Bassily et al. [2, Section C], Sekhari et al. [28] show that any unlearning algorithm that ignores the samples $U$ can't improve upon the DP lower bound. This motivates the study of algorithms specifically designed for unlearning which leverage samples from $U$ instead of algorithms based on DP. In Appendix B, we provide details showing that our Algorithm 1 achieves the lower bound (7), where the constant $c$ depends on the Lipschitz constants $L$, $M$, strongly convex constant $\mu$, and boundedness constant $C$.

**Comparison to previous results**   We compare our method to Sekhari et al. [28] and Guo et al. [14] who propose optimizing a second-order Taylor approximation (**TA**) to the leave-$U$-out objective function (2) for batch removal of $U$. This results in the Newton-like removal mechanism (8).

$$\tilde{\theta}_{n,-U}(\lambda) = \hat{\theta}_n(\lambda) + \frac{1}{n-m}\left(\frac{1}{n-m}\sum_{z\in S\setminus U}\nabla_\theta^2 F(z,\hat{\theta}_n(\lambda),\lambda)\right)^{-1}\sum_{z\in U}\nabla\ell(z,\hat{\theta}_n(\lambda)). \qquad (8)$$

*Comparison of Assumptions.*   On the one hand, our Assumption 1 is slightly more restrictive than that of Sekhari et al. [28] given we require boundedness of the Hessian loss to obtain our unlearning, generalization and deletion capacity guarantees. Notably, this assumption does not rule out any of the most common convex objective functions (e.g. least squares, logistic, hinge, or cross entropy loss functions). On the other hand, our technique allows for non-smooth regularizers which are common in modern machine learning pipelines.

*Comparison of Computation.*   Algorithm 1 entails calculating and inverting the full Hessian $H_f$ or $H_\ell$ and storing it (memory cost $O(d^2)$ and computation cost $O(d^3)$). It also entails storing the gradient of each point evaluated on the full data model $\hat{\theta}_n(\lambda)$, as well as the current model $\tilde{\theta}_{n,-U}(\lambda)$. In the setting where $\pi$ is not smooth, $\bar{\theta}_{n,-U}(\lambda)$ must be stored. During runtime, Algorithm 1 requires matrix-vector multiplication and vector addition (as well as proximal step in the non-smooth setting). For unlearning in settings where deletion requests come in a online manner, our technique is computationally more efficient. This is because removal mechanism (8) requires computing and inverting a different Hessian that depends on the user requesting the deletion. Therefore, outside simple settings, complying with such requests involves a computational cost of $O(md^3)$ to remove $m$ datapoints.

*Comparison of unlearning, generalization and deletion capacity.* As summarized above, the generalization, unlearning and deletion capacity results for our removal mechanism are essentially equivalent to the unlearning, generalization, and deletion capacity results of [28].

**Remark 1** (Extending Sekhari et al. [28] to non-smooth regularizers)*. Following from the equivalence we show between the batch and online setting in Appendix A.2, we can extend our use of the proximal operator to (8) which would extend the results of Sekhari et al. [28] to non-smooth regularizers. Similar unlearning, deletion capacity and generalization results are obtained. We provide details in Appendix C.*

# 4   Experiments

In this section we will refer to retraining from scratch as **RT**, the Algorithm (8) as **TA** and our Algorithm 1 as **IJ**. We empirically demonstrate the benefits of our algorithm over both **RT** and **TA** in three different settings: (i) smooth regularizers where we train a logistic regression model with an $\ell_2$ penalty to predict between the digit 3 and 8 from the MNIST dataset [19], (ii) non-smooth regularizers where we train a logistic regression model with an $\ell_1$ penalty to predict whether an individual was prescribed a Warfarin dosage of $> 30$ mg/week from a dataset released by the International Warfarin Pharmacogenetics Consortium [8], and (iii) non-convex training where we apply a logistic regression model with an $\ell_2$ penalty to predict street digits signs 3 and 8 from SVHN [24] on representations extracted from a differentially private feature extractor with $\epsilon = 0.1$ (similar to the setup of Guo et al. [14]). For all algorithms, we tune $\lambda$ over the set $\{10^{-3}, 10^{-4}, 10^{-5}, 10^{-6}\}$. In the Appendix D, we provide further details about each dataset and the code to produce our models is attached separately.

We present results in the main paper for $\lambda = 1^{-3}$ and provide other results in Appendix E. All models are trained on a single NVIDIA Tesla T4 GPU and 16 2.10GHz Xeon(R) Silver 4110 CPU cores. We focus our evaluation on total runtime in seconds and test set accuracy as the number of delete requests increases. Given that the current window for complying with delete requests for GDPR is one month, the runtime savings on our plots occur at month X where X is provided by the y-axis. We note that the "Right to Be Forgotten" is a much broader right and our experimental findings demonstrate that this can be satisfied much more efficiently for convex problems. For all results, we provide the average over three different runs and provide standard error bars.

**Logistic Regression with Smooth Regularizers**   In this experiment we simulate an online data deletion setup using the MNIST dataset. For simplicity, we restrict the problem to binary classification by predicting between 3s and 8s. This is the same setup used by Guo et al. [14] to evaluate their

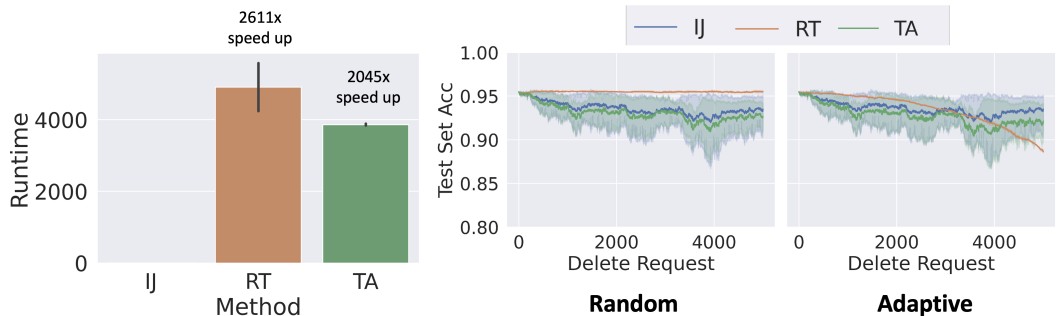

**Figure 1: IJ** vs. **RT** & **TA** for smooth regularizers. Comparing both the runtime and test accuracy of the unlearned models in our $\ell_2$ logistic regression setup for predicting 3's and 8's on MNIST. For GDPR, the y-axis of the runtime graph denotes the runtime at month X under the current one month delete request window.

approximate unlearning algorithm. We flatten the MNIST image into a 1-D vector and train an $\ell_2$ regularized logistic regression model for each algorithm. We evaluate the impact of a deleting a sequence of 5000 datapoints (approximately 40% of the total dataset) both randomly and in an adaptively chosen manner. In these experiments we consider noise at $c = 0.01$.

On average, **IJ** was 2611x faster than **RT** and 2045x faster than **TA** (Figure 1). We find that this improvement in runtime comes at very minimal cost to the test performance of the model returned by our algorithm compared to the test performance of **TA** (Figure 1). Furthermore, we observe that these results also hold when the delete requests are chosen adaptively (Figure 1). As discussed previously, the main savings in computation for our algorithm comes from only inverting the hessian once while **TA** requires a hessian inversion for every delete request.

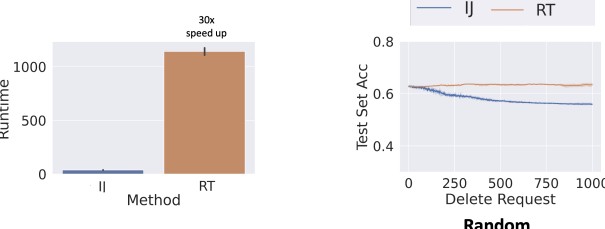

**Figure 2: IJ** vs. **RT** for non-smooth regularizers. Comparing both the runtime and test accuracy of the unlearned models in our $\ell_1$ logistic regression setup for predicting warfarin dosage. For GDPR, the y-axis of the runtime graph denotes the runtime at month X under the current one month delete request window.

**Logistic Regression with Non-Smooth Regularizers**   This experiment showcases the performance of our proximal Newton algorithm on a problem with a non-smooth regularizer. We focus on predicting warfarin dosing from patient demographic and physiological data because it a practical setting where $\ell_1$ regularization has been demonstrated to be preferred [30]. Given that **TA** does not naturally support non-smooth regularizers we only compare **IJ** to **RT**. In these experiments we consider a logistic regression objective, $\ell_1$ regularizer, and noise level of $c = 0.01$. On average, **IJ** was 30x faster than **RT** (Figure 2). This improvement in runtime only comes at small cost to the test accuracy of the model returned by our algorithm compared to the test performance of **TA** (Figure 2).

**Non-Convex: Logistic Regression with Differentially Private Feature Extractor**   We demonstrate the ability to use our unlearning algorithm in non-convex settings. Similar to Guo et al. [14] we train a differentially private feature extractor with $\epsilon = 0.1$ on street digit signs from SVHN to extract representations which a logistic regression model can be applied on top of. Similar to previous experiments, we pick digits 3 and 8 for simplicity and observe significant speedups using **IJ**. On average, **IJ** was 15036x faster than **RT** and 17x faster than **TA** and these speedups come at a marginal cost to the test accuracy of our algorithm even as the number of delete requests increase (Figure 3).

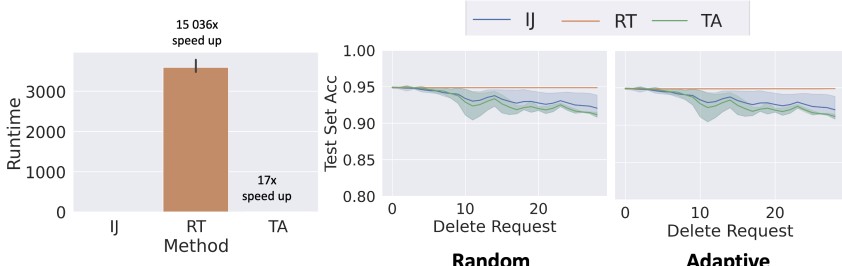

**Figure 3: IJ** vs. **TA** and **RT** for non-convex training. Comparing both the runtime and test accuracy of the unlearned models in our DP feature extractor + $\ell_2$ logistic regression setup for predicting digits in photos of street signs from SVHN. For GDPR, the y-axis of the runtime graph denotes the runtime at month X under the current one month delete request window.

## 5 Unlearning hyperparameter-tuned models

One of the most common techniques for hyperparameter tuning is *cross-validation*. Here datapoints are used to validate and select models using the following cross-validation error objective

$$\text{CV}(\lambda) = \frac{1}{n} \sum_{i=1}^{n} \ell(z_i, \hat{\theta}_{n,-i}(\lambda)). \tag{9}$$

Specifically, the model selection pipeline oftentimes entails selecting $\lambda \in \Lambda$ that minimizes the CV error (9). Given each datapoint is used to select models, one might hope it is possible to unlearn models using Algorithm 1 when CV is used to select the model. However, as the following proposition illustrates, the $(\epsilon, \delta)$-unlearning guarantees of Theorem 1 no longer apply when CV has taken place.

**Proposition 1.** *Suppose cross-validation is used to select $\lambda \in \Lambda$ and the empirical risk minimizer $\hat{\theta}_n(\lambda)$ is returned. Consider a delete request by user $i$ and the model returned by unlearning procedure (8) or (1), which we denote $\tilde{\theta}_{n,-i}(\lambda)$. Then it is possible $\|\tilde{\theta}_{n,-i}(\lambda) - \hat{\theta}_{n,-i}(\lambda')\| = o(1/n)$ where $\hat{\theta}_{n,-i}(\lambda')$ is the model selected after deleting user $i$ and performing cross-validation again to select $\lambda' \in \Lambda$ before returning the empricial risk minizer.*

*Proof of Proposition 1.* Suppose $\ell(z, \theta) = \frac{1}{2}(z - \theta)^2$, $\pi(\theta) = \theta^2$ and $\Lambda = \{0, \infty\}$. Note that $\hat{\theta}_n(\lambda) = \frac{1}{1+\lambda}\bar{z}$ where $\bar{z} = \frac{1}{n}\sum_{i=1}^{n} z_i$ is the sample average. Consider a dataset consisting of $n - 1$ points (group A) with value $-\frac{1}{n}$ and a remaining point (group B) with value $n$ with $n \geq 2$, and suppose the user in group B requests to delete their data. Performing cross-validation on this dataset will entail computing two kinds of sample averages, one where we have deleted a point from group A, $\bar{z}_{-i_A} = \frac{1}{n-1}(-\frac{n-2}{n} + n)$, and one where we have deleted the point from group B, $\bar{z}_{-i_B} = -\frac{1}{n}$. Recall the CV error

$$\text{CV}(\lambda) = \frac{1}{2n}\sum_{i=1}^{n}(z_i - \hat{\theta}_{n,-i}(\lambda))^2 = \frac{n-1}{2n}\left(\frac{1}{1+\lambda}\frac{1}{n-1}(n - \frac{n-2}{n}) - \frac{1}{n}\right)^2 + \frac{1}{2n}(n + \frac{1}{1+\lambda}\frac{1}{n})^2$$

When minimized over the set $\Lambda = \{0, \infty\}$, $\lambda = \infty$ is optimal for any $n \geq 2$. This results in estimator $\hat{\theta}_n(\lambda) = \hat{\theta}_n(\infty) = 0$. The leave-one-out approximation is the same for estimators (1) and (8), given $\nabla_\theta^2 F(z, \theta, \lambda)^{-1} = \frac{1}{1+\lambda}$. Subsequently, $\tilde{\theta}_{n,-i}(\lambda) = \hat{\theta}_n(\lambda) + \frac{1}{n}\frac{1}{1+\lambda}(\hat{\theta}_n(\lambda) - z_i)$ results in the approximation $\tilde{\theta}_{n,-i}(\infty) = 0$. On the other hand, having deleted the datapoint from group $B$, we have a new sample mean $\bar{z} = -\frac{1}{n}$ and CV error, given by

$$\text{CV}(\lambda) = \frac{1}{2n}\sum_{i=1}^{n}(z_i - \hat{\theta}_{n,-i}(\lambda))^2 = \frac{1}{2n}\left(-\frac{1}{n} + \frac{1}{1+\lambda}\frac{1}{n}\right)^2$$

is minimized by $\lambda = 0$. This results in leave-one-out minimizer $\hat{\theta}_{n,-i}(0) = -\frac{1}{n}$. $\square$

An implication of Proposition 1 is that Algorithm 1 as well as the algorithms proposed by [28, 14] do not unlearn models when cross-validation is used to selected hyperparameters. This is because the noise term $\sigma \propto O(m^2/n^2)$ is no longer sufficient to guarantee datapoint $i$ is unlearned. This means that organizations which train their models using cross-validation and use approximate unlearning algorithms could still be leaking information about the data which they delete.

# 6 Discussion

A main contribution of our study is the development of an efficient online/batch data deletion algorithm for non-smooth problems. The only previous work proposing a low-memory online algorithm is Guo et al. [14]. However, Marchant et al. [20] show that this algorithm is susceptible to poisoning attacks rendering it highly inefficient. While exact unlearning algorithms such as SISA and retraining naturally work in an online setting, our work is the first to make this work for approximate unlearning algorithms. Further development of data deletion algorithms should focus more minimizing computational cost on the streaming batch setting to ensure practical use. Also, while the current form of GDPR legislation allows quite a bit of time for companies to comply (30 days), we believe that showing a tool can provide the same empirical performance and theoretical guarantees with immediate deletion is beneficial toward encouraging companies to complying with GDPR requests more swiftly (and might encourage lawmakers to necessitate faster compliance). When individuals request to delete their data, it is sometimes because they are concerned about the risk of potential harm if their data remains available; this risk is potentially compounded the longer it takes for the data to be deleted. By showing it can be done quickly, companies may be encouraged to act more expeditiously and less harm might occur.

The infinitesimal jacknife has been previously been used to perform cross-validation. We note deep connections between the approximate cross-validation algorithms and approximate machine unlearning algorithms. The algorithm developed by [28], for example, can be viewed as an analog of the approximate cross-validation algorithm proposed by [3, 26]. Additionally, the model selection error bounds provided in the approximate cross-validation literature for example in [35] are very similar to the generalization guarantees proved in our work and [28] where we are concerned with the error introduced by our approximation to the unlearning baseline. We believe further connections could be exploited to provide unlearning algorithms when hyperparameter tuning has taken place.

In this work and in several others we consider a definition of data deletion that is parameterized by two values $\epsilon$ and $\delta$. As is the case with differential privacy, it is currently unclear what the impact of satisfying different levels of $\epsilon$ means practically. Developing auditing algorithms [32] similar to those recently seen in the DP community [18, 22] can help provide more transparency on the meaning of the $(\epsilon, \delta)$-unlearning guarantee.

Finally, we return to the broader question we posed at the beginning: *what does it mean to delete data from a machine learning pipeline?* Most existing work focuses on data deletion in the model training process. Yet, the machine learning and data analysis pipeline is much broader than just model training. As evidenced in Section 5, the lack of research on data deletion in model selection could potentially result in information being leaked from previously proposed unlearning algorithms. Given this result, it is likely that all unlearning algorithms (*both exact and inexact*) still leak information about deleted data in other stages of the machine learning pipeline such as exploratory data analysis and feature selection. We encourage the community to explore definitions of data deletion which encompass the entire machine learning pipeline.

# 7 Acknowledgements

The authors would like to thank the NeurIPS 2022 reviewers and area chair for their comments and feedback. The authors acknowledge the MIT SuperCloud and Lincoln Laboratory Supercomputing Center for providing (HPC, database, consultation) resources that have contributed to the research results reported within this paper/report. VMS is supported by a Wellcome Trust Fellowship.

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
