# A Proof of (1)

**Lemma 2** (Optimization comparison lemma [35]). *Suppose*

$$x^* \in \operatorname*{argmin}_x \varphi_1(x) + \varphi_0(x) \quad and \quad y^* \in \operatorname*{argmin}_x \varphi_2(x) + \varphi_0(x). \tag{10}$$

*for $\varphi_1$ and $\varphi_2$ differentiable and $\varphi_0$ convex.*

*Proof.* The (sub)differentiability assumptions and the optimality of $x_{\varphi_1}$ and $x_{\varphi_2}$ imply that $0 \in \partial \varphi_2$ and $u = 0 + \nabla(\varphi_1 - \varphi_2)(x_{\varphi_1})$ for some $u \in x_{\varphi_2}$. The gradient growth condition implies

$$\nu_{\varphi_2}(\|x_{\varphi_1} - x_{\varphi_2}\|_2) \leq \langle x_{\varphi_1} - x_{\varphi_2}, u - 0 \rangle = \langle x_{\varphi_1} - x_{\varphi_2}, \nabla(\varphi_2 - \varphi_1)(x_{\varphi_1}) \rangle. \tag{11}$$

$\square$

**Lemma 3** (Learning guarantee for $\hat{\theta}_n(\lambda)$). *Given, $F_n$ satisfies Assumption 1 or 2 and any distribution $\mathcal{D}$, let $S = \{z_i\}_{i=1}^n$ where $S \sim \mathcal{D}^n$. Then the the empirical minimizer $\hat{\theta}_n(\lambda)$ of $F_n(\theta, \lambda, z)$ satisfies*

$$\mathbb{E}[F(\hat{\theta}_n(\lambda)) - F(\theta^*(\lambda))] \leq \frac{4L^2}{\mu n}$$

*Proof.* Given $F_n$ is $\mu$-strongly convex this follows from Claim 6.2 in [29]. $\square$

## A.1 Proof of (6b): Closeness of $\hat{\theta}_n(\lambda)$ and $\hat{\theta}_{n,-U}(\lambda)$

Suppose we have deleted $m$ users in a set $U$. Define $\tilde{F}_{n,-U} = \frac{n-m}{n} F_{n,-U}$ where $F_{n,-U} = \frac{1}{n-m} \sum_{i \notin U} f(z_i, \theta, \lambda)$ and note that $\tilde{F}_{n,-U}$ and $F_{n,-U}$ have the same minimizers. We will work with $\tilde{F}_{n,-U}$. By the optimizer comparison lemma 2 and strong convexity of $F_n$

$$
\begin{aligned}
\mu \|\hat{\theta}_n(\lambda) - \hat{\theta}_{n,-U}(\lambda)\|_2^2 &\leq \langle \hat{\theta}_n(\lambda) - \hat{\theta}_{n,-U}(\lambda), \nabla F_n(z, \hat{\theta}_n(\lambda), \lambda) - \nabla \tilde{F}_{n,-U}(z, \hat{\theta}_n(\lambda), \lambda) \rangle \\
&= \frac{1}{n} \sum_{i \in U} \langle \hat{\theta}_n(\lambda) - \hat{\theta}_{n,-U}(\lambda), \nabla \ell(z_i, \hat{\theta}_n(\lambda)) \rangle \\
&\leq \frac{1}{n} \|\hat{\theta}_n(\lambda) - \hat{\theta}_{n,-U}(\lambda)\|_2 \sum_{i \in U} \|\nabla \ell(z_i, \hat{\theta}_n(\lambda))\|_2 \\
&\leq \frac{1}{n} \|\hat{\theta}_n(\lambda) - \hat{\theta}_{n,-U}(\lambda)\|_2 \cdot mL
\end{aligned}
$$

Dividing both sides by $\|\hat{\theta}_n(\lambda) - \hat{\theta}_{n,-U}(\lambda)\|_2$ and rearranging gives the desired bound of

$$\|\hat{\theta}_n(\lambda) - \hat{\theta}_{n,-U}(\lambda)\|_2 \leq \frac{mL}{\mu n}$$

### A.1.1 Proof of (6b): Closeness of $\hat{\theta}_{n,-U}(\lambda)$ and $\bar{\theta}_{n,-U}(\lambda)$

We define:

- $\psi_1 = \tilde{F}_{n,-U}(z, \theta, \lambda)$
- $\psi_2 = \langle \nabla \tilde{F}_{n,-U}(\hat{\theta}_n(\lambda)), \hat{\theta}_n(\lambda) - \theta \rangle + \langle \hat{\theta}_n(\lambda) - \theta, \nabla_\theta^2 \tilde{F}_{n,-U}(\hat{\theta}_n(\lambda))[\hat{\theta}_n(\lambda) - \theta] \rangle$
- $\psi_3 = \langle \nabla \tilde{F}_{n,-U}(\hat{\theta}_n(\lambda)), \hat{\theta}_n(\lambda) - \theta \rangle + \langle \hat{\theta}_n(\lambda) - \theta, \nabla_\theta^2 \tilde{F}_n(\hat{\theta}_n(\lambda))[\hat{\theta}_n(\lambda) - \theta] \rangle$
- $\hat{\theta}_{n,-U}(\lambda) = \operatorname{argmin} \psi_1(\theta)$,
- $\tilde{\theta}_{n,-U}(\lambda) = \operatorname{argmin} \psi_3(\theta)$

The optimizer comparison theorem and strong convexity of $F_n$ implies the following upper bound:

$$
\begin{aligned}
\frac{\mu}{2} \|\hat{\theta}_{n,-U}(\lambda) - \tilde{\theta}_{n,-U}(\lambda)\|_2^2 &\leq \langle \hat{\theta}_{n,-U}(\lambda) - \tilde{\theta}_{n,-U}(\lambda), \nabla(\psi_3 - \psi_1)(\hat{\theta}_{n,-U}(\lambda)) \rangle \\
&\leq \|\hat{\theta}_{n,-U}(\lambda) - \tilde{\theta}_{n,-U}(\lambda)\|_2 \|\nabla(\psi_3 - \psi_1)(\hat{\theta}_{n,-U}(\lambda))\|_2
\end{aligned}
$$

Dividing both sides by $\|\hat{\theta}_{n,-U}(\lambda) - \tilde{\theta}_{n,-U}(\lambda)\|_2$ gives

$$
\begin{aligned}
\frac{\mu}{2}\|\hat{\theta}_{n,-U}(\lambda) - \tilde{\theta}_{n,-U}(\lambda)\|_2 &\leq \|\nabla_\theta(\psi_3 - \psi_2)(\hat{\theta}_{n,-U}(\lambda)) - \nabla_\theta(\psi_2 - \psi_1)(\hat{\theta}_{n,-U}(\lambda))\|_2 \\
&\leq \|\nabla_\theta(\psi_3 - \psi_2)(\hat{\theta}_{n,-U}(\lambda))\|_2 + \|\nabla_\theta(\psi_2 - \psi_1)(\hat{\theta}_{n,-U}(\lambda))\|_2 \\
&\leq \|\nabla_\theta^2 \tilde{F}_n(\hat{\theta}_{n,-U}(\lambda)) - \nabla_\theta^2 \tilde{F}_{n,-U}(\hat{\theta}_{n,-U}(\lambda))\|_2 \|\hat{\theta}_n(\lambda) - \hat{\theta}_{n,-U}(\lambda)\|_2 \\
&\quad + \|\nabla_\theta(\psi_2 - \psi_1)(\hat{\theta}_{n,-U}(\lambda))\|_2 \\
&\leq \frac{m^2 CL}{\mu n^2} + \|\nabla_\theta \psi_2(\hat{\theta}_{n,-U}(\lambda)) - \nabla_\theta \psi_1(\hat{\theta}_{n,-U}(\lambda))\|_2 \\
&\overset{①}{\leq} \frac{m^2 CL}{\mu n^2} + \frac{M}{2}\|\hat{\theta}_{n,-U}(\lambda) - \hat{\theta}_n(\lambda)\|_2^2 \\
&\leq \frac{m^2 CL}{\mu n^2} + \frac{M}{2}\cdot\frac{m^2 L^2}{\mu^2 n^2}
\end{aligned}
$$

Inequality ① follows from smoothness of the objective function. Dividing both sides by $\frac{\mu}{2}$, gives the desired bound of

$$
\|\hat{\theta}_{n,-U}(\lambda) - \tilde{\theta}_{n,-U}(\lambda)\|_2 \leq \frac{2m^2 CL}{\mu^2 n^2} + \frac{Mm^2 L^2}{\mu^3 n^2}
$$

For the non-smooth version of our algorithm, the same proof holds where we define

- $\psi_1 = \tilde{\ell}_{n,-U}(z, \theta, \lambda)$
- $\psi_2 = \langle \nabla\tilde{\ell}_{n,-U}(\hat{\theta}_n(\lambda)), \hat{\theta}_n(\lambda) - \theta \rangle + \langle \hat{\theta}_n(\lambda) - \theta, \nabla_\theta^2 \tilde{\ell}_{n,-U}(\hat{\theta}_n(\lambda))[\hat{\theta}_n(\lambda) - \theta]\rangle + \pi(\theta)$
- $\psi_3 = \langle \nabla\tilde{\ell}_{n,-U}(\hat{\theta}_n(\lambda)), \hat{\theta}_n(\lambda) - \theta \rangle + \langle \hat{\theta}_n(\lambda) - \theta, \nabla_\theta^2 \tilde{\ell}_n(\hat{\theta}_n(\lambda))[\hat{\theta}_n(\lambda) - \theta]\rangle + \pi(\theta)$
- $\hat{\theta}_{n,-U}(\lambda) = \arg\min \psi_1(\theta)$,
- $\tilde{\theta}_{n,-U}(\lambda) = \arg\min \psi_3(\theta)$

$$
\begin{aligned}
\frac{\mu}{2}\|\hat{\theta}_{n,-U}(\lambda) - \tilde{\theta}_{n,-U}(\lambda)\|_2 &\leq \|\nabla_\theta^2 \tilde{\ell}_n(\hat{\theta}_{n,-U}(\lambda)) - \nabla_\theta^2 \tilde{\ell}_{n,-U}(\hat{\theta}_{n,-U}(\lambda))\|_2 \|\hat{\theta}_n(\lambda) - \hat{\theta}_{n,-U}(\lambda)\|_2 \\
&\quad + \|\nabla_\theta \psi_2(\hat{\theta}_{n,-U}(\lambda)) - \nabla_\theta \psi_1(\hat{\theta}_{n,-U}(\lambda))\|_2 \\
&\leq \frac{m^2 CL}{\mu n^2} + \|\nabla_\theta \psi_2(\hat{\theta}_{n,-U}(\lambda)) - \nabla_\theta \psi_1(\hat{\theta}_{n,-U}(\lambda))\|_2 \\
&\leq \frac{m^2 CL}{\mu n^2} + \frac{M}{2}\|\hat{\theta}_{n,-U}(\lambda) - \hat{\theta}_n(\lambda)\|_2^2 \\
&\leq \frac{m^2 CL}{\mu n^2} + \frac{M}{2}\cdot\frac{m^2 L^2}{\mu^2 n^2}
\end{aligned}
$$

## A.2 Comparisons between batch and streaming algorithm

We show that the batch and streaming version of the algorithms are equivalent.

**Case 1: $\pi$ is smooth.** The bounds we have proved are for the minizmier of $\varphi_3$, namely

$$
\begin{aligned}
\tilde{\theta}_{n,-U}(\lambda) &= \hat{\theta}_n(\lambda) - \nabla_\theta^2 \tilde{F}(\hat{\theta}_n(\lambda))^{-1}\nabla \tilde{F}_{n,-U}(\hat{\theta}_n(\lambda)) \\
&= \hat{\theta}_n(\lambda) + \frac{1}{n}\left(\frac{1}{n}\sum_{i=1}^n \nabla_\theta^2 F(z_i, \hat{\theta}_n(\lambda), \lambda)\right)^{-1}\sum_{i\in U}\nabla\ell(z_i, \hat{\theta}_n(\lambda))
\end{aligned}
$$

Now suppose 1 datapoint (user $j$) requests to be deleted. Then the streaming and batch algorithms agree, as the update becomes

$$
\tilde{\theta}_{n,-i}(\lambda) = \hat{\theta}_n(\lambda) + \frac{1}{n}\left(\frac{1}{n}\sum_{i=1}^n \nabla_\theta^2 F(z_i, \hat{\theta}_n(\lambda), \lambda)\right)^{-1}\nabla\ell(z_i, \hat{\theta}_n(\lambda)).
$$

Now suppose the algorithms are consistent for all deletion requests in the set $U$. When an additional user $j$ requests to delete their data the streaming algorithm returns

$$
\begin{aligned}
\tilde{\theta}_{n,-(U\cup\{j\})}(\lambda) &= \tilde{\theta}_{n,-U}(\lambda) + \frac{1}{n}\left(\frac{1}{n}\sum_{i=1}^n \nabla_\theta^2 F(z_i, \hat{\theta}_n(\lambda), \lambda)\right)^{-1}\nabla\ell(z_j, \hat{\theta}_n(\lambda)) \\
&= \hat{\theta}_n(\lambda) + \frac{1}{n}\left(\frac{1}{n}\sum_{i=1}^n \nabla_\theta^2 F(z_i, \hat{\theta}_n(\lambda), \lambda)\right)^{-1}\sum_{i\in U}\nabla\ell(z_i, \hat{\theta}_n(\lambda)) \\
&\quad + \frac{1}{n}\left(\frac{1}{n}\sum_{i=1}^n \nabla_\theta^2 F(z_i, \hat{\theta}_n(\lambda), \lambda)\right)^{-1}\nabla\ell(z_j, \hat{\theta}_n(\lambda)) \\
&= \hat{\theta}_n(\lambda) + \frac{1}{n}\left(\frac{1}{n}\sum_{i=1}^n \nabla_\theta^2 F(z_i, \hat{\theta}_n(\lambda), \lambda)\right)^{-1}\nabla\sum_{i\in(U\cup\{j\})}\ell(z_i, \hat{\theta}_n(\lambda))
\end{aligned}
$$

which matches the batch version of the deletion algorithm. This inductive arguments show both batch and streaming algorithms are the same.

**Case 2: $\pi$ is not smooth.** When $\pi$ is not smooth, the minimizer of $\varphi_3$ satisfies

$$\tilde{\theta}_{n,-(U\cup\{j\})}(\lambda) = \tilde{\theta}_{n,-U}(\lambda) + \tfrac{1}{n}(\tfrac{1}{n}\sum_{i=1}^{n}\nabla_\theta^2 F(z_i,\hat{\theta}_n(\lambda),\lambda))^{-1}\nabla\ell(z_j,\hat{\theta}_n(\lambda)) + \lambda\nabla\pi(\tilde{\theta}_{n,-(U\cup\{j\})}(\lambda))$$

When 1 datapoint (user $j$) requests to be deleted, the streaming and batch algorithms agree given $U = \emptyset$. Now suppose the algorithms are consistent for all deletion requests in the set $U$. When an additional user $j$ requests to delete their data the streaming algorithm returns an estimator that satisfies

$$\begin{aligned}
\bar{\theta}_{n,-(U\cup\{j\})}(\lambda) &= \bar{\theta}_{n,-U}(\lambda) + \tfrac{1}{n}H_\ell^{-1}\nabla\ell(z_j,\hat{\theta}_n(\lambda)) + \lambda H_\ell^{-1}\nabla(\bar{\theta}_{n,-(U\cup\{j\})}(\lambda)) \\
&= \hat{\theta}_n(\lambda) + \tfrac{1}{n}H_\ell^{-1}\nabla\sum_{i\in(U\cup\{j\})}\ell(z_i,\hat{\theta}_n(\lambda)) + \lambda H_\ell^{-1}\nabla(\bar{\theta}_{n,-(U\cup\{j\})}(\lambda))
\end{aligned}$$

which matches the batch version of the deletion algorithm. This inductive arguments show both batch and streaming algorithms are the same.

### A.3 Proof of excess empirical risk

Second, we prove the excess empirical risk of our unlearning algorithm (1).

*Proof.*

$$\begin{aligned}
\mathbb{E}[F_n(\tilde{\theta}_{n,-U}(\lambda)) - F_n(\theta^*(\lambda))] &= \mathbb{E}[F_n(\tilde{\theta}_{n,-U}(\lambda)) - F_n(\hat{\theta}_n(\lambda)) + F_n(\hat{\theta}_n(\lambda)) - F_n(\theta^*(\lambda))] \\
&= \mathbb{E}[F_n(\tilde{\theta}_{n,-U}(\lambda)) - F_n(\hat{\theta}_n(\lambda))] + \mathbb{E}[F_n(\hat{\theta}_n(\lambda)) - F_n(\theta^*(\lambda))] \\
&\overset{①}{\leq} \mathbb{E}[L\|\tilde{\theta}_{n,-U}(\lambda) - \hat{\theta}_n(\lambda)\|] + \tfrac{4L^2}{\mu n}
\end{aligned}$$

where ① comes from Lemma 3 given that $F_n$ satisfies Assumption 1 or 2.

Next we upper bound $\mathbb{E}[\|\tilde{\theta}_{n,-U}(\lambda) - \hat{\theta}_n(\lambda)\|]$:

$$\begin{aligned}
\mathbb{E}[\|\tilde{\theta}_{n,-U}(\lambda) - \hat{\theta}_n(\lambda)\|] &= \mathbb{E}[\|\tilde{\theta}_{n,-U}(\lambda) - \hat{\theta}_{n,-U}(\lambda) + \hat{\theta}_{n,-U}(\lambda) - \hat{\theta}_n(\lambda)\|] \\
&= \mathbb{E}[\|\tilde{\theta}_{n,-U}(\lambda) - \hat{\theta}_{n,-U}(\lambda)\|] + \mathbb{E}[\|\hat{\theta}_{n,-U}(\lambda) - \hat{\theta}_n(\lambda)\|] \\
&\overset{②}{\leq} \mathbb{E}[\|\tilde{\theta}_{n,-U}(\lambda) - \hat{\theta}_{n,-U}(\lambda)\|] + \tfrac{mL}{\mu n} \\
&\leq \mathbb{E}[\|\bar{\theta}_{n,-U}(\lambda) - \hat{\theta}_{n,-U}(\lambda) + \sigma\|] + \tfrac{mL}{\mu n} \\
&\leq \mathbb{E}[\|\bar{\theta}_{n,-U}(\lambda) - \hat{\theta}_{n,-U}(\lambda)\|] + \mathbb{E}[\|\sigma\|] + \tfrac{mL}{\mu n} \\
&\overset{③}{\leq} \tfrac{2m^2 CL}{\mu^2 n^2} + \tfrac{Mm^2 L^2}{\mu^3 n^2} + \sqrt{d}c + \tfrac{mL}{\mu n} \\
&\leq \tfrac{2m^2 CL}{\mu^2 n^2} + \tfrac{Mm^2 L^2}{\mu^3 n^2} + \tfrac{\sqrt{d}\sqrt{2\ln(1.25/\delta)}}{\epsilon}\left(\tfrac{2m^2 CL}{\mu^2 n^2} + \tfrac{Mm^2 L^2}{\mu^3 n^2}\right) + \tfrac{mL}{\mu n}
\end{aligned}$$

where ② comes from Lemma 1 and ③ comes from Jensen's inequality and Lemma 1 (Equation 6b).

Now we substitute this back into our earlier bound:

$$\begin{aligned}
\mathbb{E}[F_n(\tilde{\theta}_{n,-U}(\lambda)) - F_n(\theta^*(\lambda))] &\leq L\left(\tfrac{2m^2 CL}{\mu^2 n^2} + \tfrac{Mm^2 L^2}{\mu^3 n^2} + \tfrac{\sqrt{d}\sqrt{2\ln(1.25/\delta)}}{\epsilon}\left(\tfrac{2m^2 CL}{\mu^2 n^2} + \tfrac{Mm^2 L^2}{\mu^3 n^2}\right) + \tfrac{mL}{\mu n}\right) + \tfrac{4L^2}{\mu n} \\
&\leq \tfrac{2m^2 CL^2}{\mu^2 n^2} + \tfrac{Mm^2 L^3}{\mu^3 n^2} + \tfrac{\sqrt{d}\sqrt{2\ln(1.25/\delta)}}{\epsilon}\left(\tfrac{2m^2 CL^2}{\mu^2 n^2} + \tfrac{Mm^2 L^3}{\mu^3 n^2}\right) + \tfrac{mL^2}{\mu n}\right) + \tfrac{4L^2}{\mu n} \\
&\leq \left(1 + \tfrac{\sqrt{d}\sqrt{2\ln(1.25/\delta)}}{\epsilon}\right)\left(\tfrac{2m^2 CL^2}{\mu^2 n^2} + \tfrac{Mm^2 L^3}{\mu^3 n^2}\right) + \tfrac{4mL^2}{\mu n} \\
&\leq \left(1 + \tfrac{\sqrt{d}\sqrt{2\ln(1.25/\delta)}}{\epsilon}\right)\left(\tfrac{(2C\mu+ML)m^2 L^2}{\mu^3 n^2}\right) + \tfrac{4mL^2}{\mu n}
\end{aligned}$$

$\square$

Finally, we prove that our unlearning algorithm (1) results in $(\epsilon, \delta)$-certifiable removal of datapoint $\mathbf{z} \in U \subseteq S$.

*Proof.* We use a similar technique to the proof of the differential privacy guarantee for the Gaussian mechanism ([9]).

Let $\hat{\theta}_n(\lambda)$ be the output of learning algorithm $A$ trained on dataset $S$ and $\tilde{\theta}_{n,-U}(\lambda)$ be the output of unlearning algorithm $M$ run on the sequence of delete requests $U$, $\hat{\theta}_n(\lambda)$, and the data statistics $T(S)$. We also note the output of $M$ before adding noise as $\bar{\theta}_{n,-U}(\lambda)$. Finally, we denote $\hat{\theta}_{n,-U}(\lambda)$ as the output of $A$ trained on the dataset $S \backslash U$.

We note that in Algorithm 1 that $\tilde{\theta}_{n,-U}(\lambda)$ is simply $\tilde{\theta}_{n,-U}(\lambda) = \bar{\theta}_{n,-U}(\lambda) + \sigma$. The noise $\sigma$ is sampled from $\mathcal{N}(0, c^2 I)$ with $c = \|\hat{\theta}_{n,-U}(\lambda) - \bar{\theta}_{n,-U}(\lambda)\|_2 \cdot \frac{\sqrt{2\ln(1.25/\delta)}}{\epsilon}$. Where $\|\hat{\theta}_{n,-U}(\lambda) - \bar{\theta}_{n,-U}(\lambda)\|_2 \leq \frac{2m^2 CL}{n^2 \mu^2} + \frac{m^2 ML^2}{n^2 \mu^3}$ (6b). Following the same proof for the DP gaurantee of the Gaussian mechanism as Dwork et al. [9] (Theorem A.1) given the noise is sampled from the previously described Gaussian distribution we get for any $\Theta$:

$$P(\hat{\theta}_{n,-U} \in \Theta) \leq e^\epsilon P(\tilde{\theta}_{n,-U} \in \Theta) + \delta, \quad \text{and}$$
$$P(\tilde{\theta}_{n,-U} \in \Theta) \leq e^\epsilon P(\hat{\theta}_{n,-U} \in \Theta) + \delta$$

resulting in $(\epsilon, \delta)$-unlearning.

$\square$

# B   Proof of Algorithm 1 Deletion Capacity

The upper bound on the excess risk (Theorem 1) implies that we can delete at least:

$$m_{\epsilon,\delta,\gamma}^{A,M}(d, n) \geq c \cdot \frac{n\sqrt{\epsilon}}{(d\log(1/\delta))^{\frac{1}{4}}}$$

where $c$ depends on the properties of function $F(z, \theta, \lambda)$. We specifically derive the value of $c$ below by substituting our deletion capacity bound as $m$ into the empirical excess risk upper bound:

$$\mathbb{E}[F(\tilde{\theta}_{n,-U}(\lambda)) - F(\theta^*(\lambda))] = O\left(\frac{(2C\mu+ML)L^2 m^2}{\mu^3 n^2} \frac{\sqrt{d}\sqrt{ln(1/\delta)}}{\epsilon} + \frac{4mL^2}{\mu n}\right) \quad (12)$$

Plugging in the deletion capacity bound $m = c \cdot \frac{n\sqrt{\epsilon}}{(d\log(1/\delta))^{\frac{1}{4}}}$ into the excess risk bound (12) then

$$\frac{(2C\mu+ML)L^2 m^2}{\mu^3 n^2} \frac{\sqrt{d}\sqrt{ln(1/\delta)}}{\epsilon} + \frac{4mL^2}{\mu n} = \frac{c^2(2C\mu+ML)L^2}{\mu^3} + \frac{4L^2 c}{\mu} \frac{n\sqrt{\epsilon}}{(d\log(1/\delta))^{\frac{1}{4}}}$$
$$\leq c\left(\frac{c(2C\mu+ML)L^2}{\mu^3} + \frac{4L^2}{\mu}\right)$$

Therefore,

$$c \leq \gamma\left(\frac{\mu^3}{(2C\mu+ML)L^2} + \frac{\mu}{4L^2}\right) \quad \Longrightarrow \quad \mathbb{E}[F(\tilde{\theta}_{n,-U}(\lambda)) - F(\theta^*(\lambda))] \leq \gamma$$

given $c \leq 1$. Note that the third line follows from the fact that $\frac{\sqrt{\epsilon}}{(d\log(1/\delta))^{\frac{1}{4}}} \leq 1$ given $\epsilon \leq 1$ and $\delta \leq 0.005$.

# C   Extension of non-smooth regularizer to [28]

Given a function $F(z, \theta, \lambda)$ with a non-smooth regularizer $\pi(\theta)$ which satisfies Assumption 2, the algorithm from Sekhari et al. [28] can use non-smooth regularizers with the same deletion capacity,

generalization, and unlearning guarantees as Algorithm 1. This follows from fact that the removal mechanism introduced by Sekhari et al. [28] minimizes $\psi_2$ in Appendix A.1. Therefore the optimizer comparison theorem can be applied and the distnace between the estimator and the leave-U-out estimator can be upper bounded by the same terms (more precisely, we can upper bound thist distance by $\frac{m^2 M L^2}{n^2 \mu^3}$).

# D   Dataset Details

`MNIST` We consider digit classification from the MNIST dataset which contains 60000 images of digits from 1-9. We select only digits 3 and 8 to simplify the task to binary classification. We flatten the original images which are $28 \times 28$ into a a vector of 784 pixels. Additionally, we allow for either random sampling or *adaptive* sampling where the probability of sampling a 3 is set to 10% and the probability of sampling an 8 is set to 90%.

`SVHN` We consider digit recognition from street signs from the SVHN dataset which contains 60000 images of street sign images that contain digits from 1-9. We select only digits 3 and 8 to simplify the task to binary classification. We flatten the original images which are $28 \times 28$ into a a vector of 784 pixels. Additionally, we allow for either random sampling or *adaptive* sampling where the probability of sampling a 3 is set to 10% and the probability of sampling an 8 is set to 90%.

`Warfarin Dosing` Warfarin is a prescription drug used to treat symptoms stemming from blood clots (e.g. deep vein thrombosis) and to help reduce the incidence of stroke and heart attack in at-risk patients. It is an anticoagulant which inhibits blood clotting but overdosing leads to excessive bleeding. The appropriate dosage for a patient dependent on demographic and physiologic factors resulting in high variance between patients. We focus on predicting small or large dosages for patients (defined as > 30mg/week) from a dataset released by the International Warfarin Pharmacogenetics Consortium [8] which contains both demographic and physiological measurements for patients. The dataset contains 5528 examples each with 62 features.

# E   Additional Experiments

**Logistic Regression with Smooth Regularizers**   We present the test accuracy results for the remaining values of $\lambda = \{10^{-4}, 10^{-5}, 10^{-6}\}$.

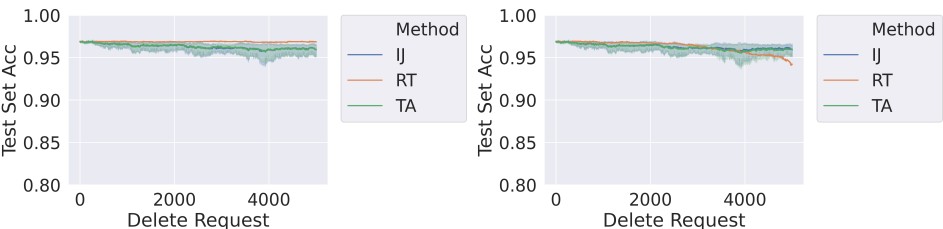

**Figure 4: IJ** vs. **RT** and **TA** for smooth regularizers. Comparing both the test accuracy of the unlearned models in our $\ell_2$ logistic regression setup for $\lambda = 10^{-4}$ for random vs adaptive sampling.

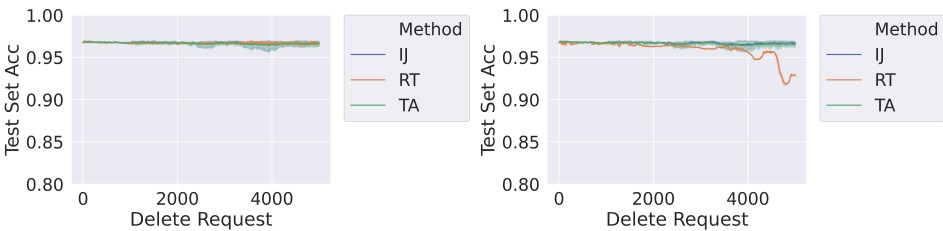

**Figure 5: IJ** vs. **RT** and **TA** for smooth regularizers. Comparing both the test accuracy of the unlearned models in our $\ell_2$ logistic regression setup for $\lambda = 10^{-5}$ for random vs adaptive sampling.

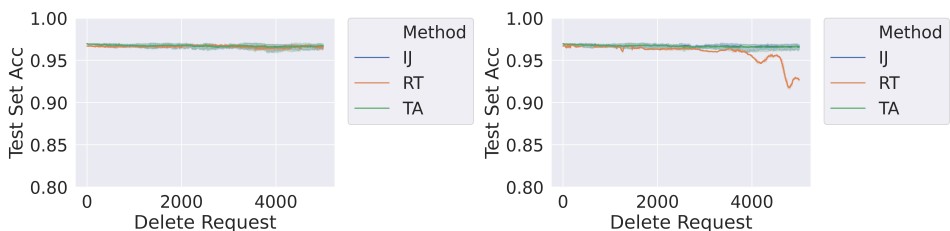

**Figure 6: IJ** vs. **RT** and **TA** for smooth regularizers. Comparing both the test accuracy of the unlearned models in our $\ell_2$ logistic regression setup for $\lambda = 10^{-6}$ for random vs adaptive sampling.

**Logistic Regression with Non-Smooth Regularizers**   We present the test accuracy results for the remaining values of $\lambda = \{10^{-4}, 10^{-5}, 10^{-6}\}$.

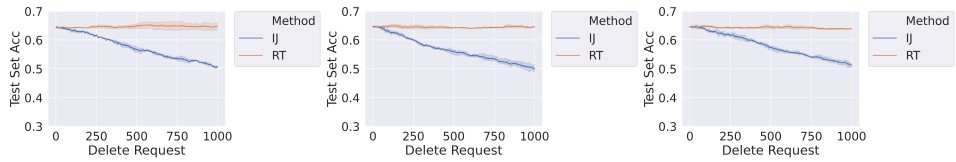

**Figure 7: IJ** vs. **RT** for non-smooth regularizers. Comparing the test accuracy of the unlearned models in our $\ell_1$ logistic regression setup for $\lambda \in \{10^{-4}, 10^{-5}, 10^{-6}\}$.

**Non-Conxex: Logistic Regression with Differentially Private Feature Extractor**   We present the test accuracy results for the remaining values of $\lambda = \{10^{-4}, 10^{-5}, 10^{-6}\}$.

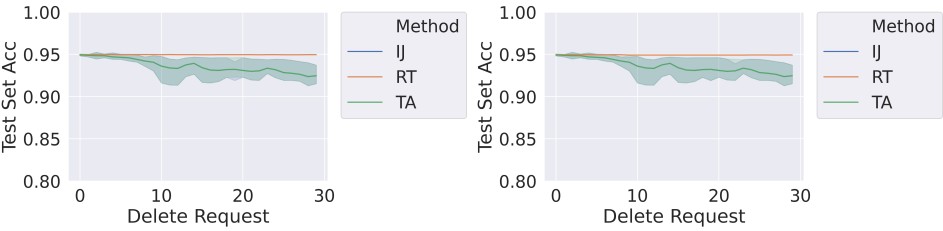

**Figure 8: IJ** vs. **TA** and **RT** for non-convex training. Comparing both the test accuracy of the unlearned models in our DP feature extractor + $\ell_2$ setup for $\lambda = 10^{-4}$.

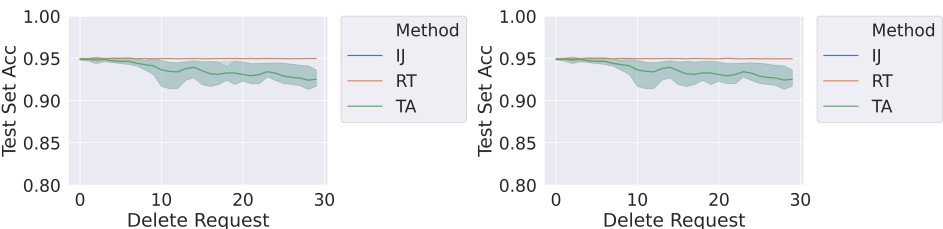

**Figure 9: IJ** vs. **TA** and **RT** for non-convex training. Comparing both the test accuracy of the unlearned models in our DP feature extractor + $\ell_2$ setup for $\lambda = 10^{-5}$.

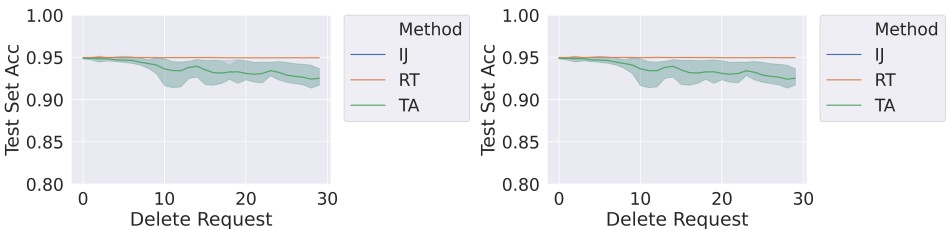

**Figure 10: IJ** vs. **TA** and **RT** for non-convex training. Comparing both the test accuracy of the unlearned models in our DP feature extractor + $\ell_2$ setup for $\lambda = 10^{-6}$.

## E.1 Runtimes

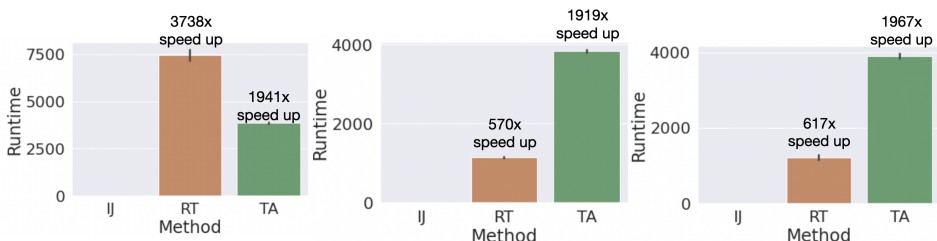

**Figure 11: IJ** vs. **RT** vs. **TA** for smooth regularizers on MNIST. Demonstrating runtime improvements across different hyperparameter settings of $10^{-4}, 10^{-5}, 10^{-6}$.

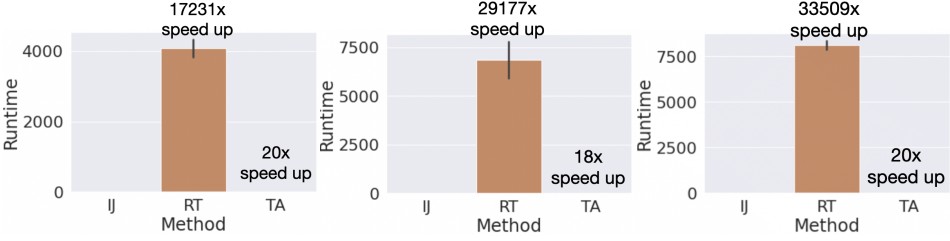

**Figure 12: IJ** vs. **RT** vs. **TA** for non-convex settings on SVHN. Demonstrating runtime improvements across different hyperparameter settings of $10^{-4}, 10^{-5}, 10^{-6}$.

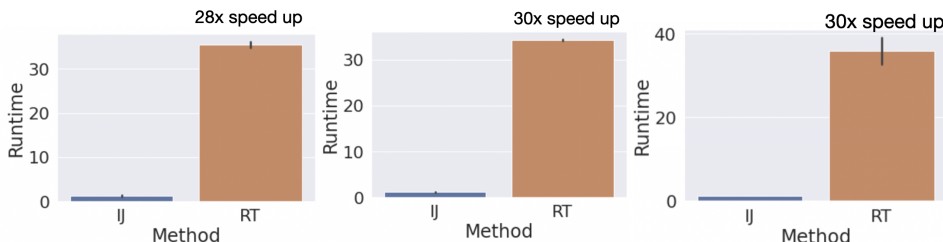

**Figure 13: IJ** vs. **RT** for non-smooth settings on Warfarin. Demonstrating runtime improvements across different hyperparameter settings of $10^{-4}, 10^{-5}, 10^{-6}$.