# OpenReview forum: "Algorithms that Approximate Data Removal: New Results and Limitations"
_NeurIPS.cc/2022/Conference — NeurIPS 2022 Accept_

### Official Review · Reviewer_EwhA · 2022-07-10

**Rating:** 6
**Confidence:** 2
**Soundness:** 3 good
**Presentation:** 3 good
**Contribution:** 3 good

**Summary:**

In this paper, the authors study the data deletion problem under the online setting. For linear models, the authors propose a second order online data deletion algorithm that enjoy better theoretical guarantees than existing methods. The algorithm can also be extended to non-smooth regularization objectives through a proximal gradient variant. Empirical studies have also shown the proposed algorithm to significantly improve over the time complexity of existing baselines while sacrificing some test accuracy. In the end, the authors propose a simple cross validation setting where all existing algorithms fail to have any theoretical guarantees.

**Questions:**

1. Could you give a concrete example where online data deletion is necessary?
2. Is there motivating examples of linear models of scale significant enough that retraining underperforms the proposed data deletion algorithm? Or alternatively, do the authors find the algorithm easily generalizable to neural networks?


**Limitations:**

I think the limitation of hyper-parameter tuning is very interesting.

On the other hand, please see the above questions regarding limitations of use cases of the proposed algorithm.

**Strengths And Weaknesses:**

Overall, I find the paper to exhibit reasonable improvement over existing work. I also find the discussion in the end of the paper regarding model tuning to be valuable and a highlight of the paper.

On the other hand, I find the setting motivation to be relatively unsupported:
1. The authors use data regulations for industrial sized models as motivation of the paper. However, most industrial models of the scale are deep neural networks. The study of only linear models seem ill-suited.
2. The authors argue that the online setting is more practical. However, taking GDPR for example, the companies are obligated to delete user data only within a month instead of immediately. As a result, I find the batched setting to be more realistic where retraining could also be competitive.
3. As a result of the above comment, is it fair to compare IJ against RT and TA in the online setting? I believe it is more reasonable compare against the methods for removing an entire batch.
4. The proposed Algorithm 1 would benefit from more detailed description and intuition.

---

> ### Author Response · Authors · 2022-08-01
> **Thank you for reviews and response to comments on motivation of unlearning in the convex setting:**
>
> **Comments on motivation of unlearning in the convex setting**
>
> **The authors use data regulations for industrial sized models as motivation of the paper. However, most industrial models of the scale are deep neural networks. The study of only linear models seem ill-suited:**
> We appreciate the reviewers concern for the applicabilty of our method to industrial sized models, specifically, neural networks. First, we point the author to several examples of industrial applications where convex (smooth and non-smooth) ERM is used. In healthcare, logistic regression models are used across the entire field for risk estimation [1]. In consumer apps such as Airbnb, convex ERM is used for optimizing pricing [2]. Our algorithm is applicable for a broad range of models generated from convex optimization problems, not just linear models.
>
> As a practical matter the IJ can be applied to neural networks, without guarantees of unlearning except in some special settings. We remark on such a setting (line 144 in our paper) where we remark it can be applied directly to non-convex models such as neural networks. The latter case was studied in [3] who also provided similar unlearning guarantees for their algorithm in this non-convex example.
>
> **Could you give a concrete example where online data deletion is necessary?** In healthcare, online data deletion is not only desireable but a requirement of some studies. For example, the UK Biobank (which is one of the largest longitudinal health data studies across over 500 000 individuals in the UK) requires data deletion as soon as a participant requests for their data to be removed [4]. In this setting, our algorithm (more broadly online data deletion) is necessary to comply with the study regulations. Importantly, many of the models trained using UK Biobank data are *not* neural networks and fall within the scope of our unlearning guarantees.
>
> We will provide these examples and more of where our algorithm can be readily applied in the revision.
>
> [1] C. Rudin and B. Ustun. Optimized scoring systems: Toward trust in machine learning for healthcare and criminal147
> justice. Interfaces, 48(5):449–466, 2018.
>
> [2] P. Ye, J. Qian, J. Chen, C.-h. Wu, Y. Zhou, S. De Mars, F. Yang, and L. Zhang. Customized regression model151
> for airbnb dynamic pricing. In Proceedings of the 24th ACM SIGKDD international conference on knowledge152
> discovery & data mining, pages 932–940, 2018
>
> [3] C. Guo, T. Goldstein, A. Hannun, and L. Van Der Maaten. Certified data removal from machine learning models.141
> In International Conference on Machine Learning, pages 3832–3842. PMLR, 2020.
>
> [4] A. Ginart, M. Guan, G. Valiant, and J. Y. Zou. Making ai forget you: Data deletion in machine learning. Advances137
> in Neural Information Processing Systems, 32, 2019

---

> > ### Author Response · Authors · 2022-08-01
> > **Comments on the usefulness of the online setting**
> >
> > **Comments on the usefulness of the online setting** While we believe we address several of these concerns in our summary response, we will answer each question specifically here.
> >
> > **The authors argue that the online setting is more practical. However, taking GDPR for example, the companies are obligated to delete user data only within a month instead of immediately. As a result, I find the batched setting to be more realistic where retraining could also be competitive:**
> >
> > We appreciate the reviewers comment on the need for an online data deletion algorithm since the current format of GDPR allows for delete requests to be performed one month after. We note that  even in the batch setting, our algorithm is less computationally intensive than previous methods after two months of complying with requests since it just needs to invert and store the Hessian once instead of every month, and therefore might *still* be preferable/useful to companies complying with monthly requests.
> > Also, while the current form of GDPR legislation allows quite a bit of time for companies to comply, we believe that showing a tool can provide the same empirical performance and theoretical guarantees with immediate deletion is beneficial toward encouraging companies to complying with GDPR requests more swiftly (and might encourage lawmakers to necessitate faster compliance). When individuals request to delete their data, it is sometimes because they are concerned about the risk of potential harm if their data remains available; this risk is compounded the longer it takes for the data to be deleted. By showing it can be done quickly, companies may be encouraged to act more expeditiously and less harm might occur.
> >
> > **As a result of the above comment, is it fair to compare IJ against RT and TA in the online setting? I believe it is more reasonable compare against the methods for removing an entire batch.**
> >
> > We prove in Appendix A.2 that the online setting of the IJ can be easily extended to a batch version. In the batch setting our method is still computationally more efficient than TA at the point where there is more than one batch request. After two months of complying with GDPR, our method will be more efficient than other low memory methods. Indeed, the relative efficiency of our method compounds the more batch delete requests there are. Thus, making our algorithm ideal for, but not restricted to, a streaming setting (this is why we chose to highlight this setting in our experiments). Furthermore, TA was not originally able to support non-smooth regularizers. Our use of the proximal operator and an equivalence we show in Remark 1 are what enables TA to now support non-smooth regularizers and perform these experiments.
> >
> > **More intuition**
> > **The proposed Algorithm 1 would benefit from more detailed description and intuition:**
> >
> > We apologize for the lack of intuition given and will provide the following intuition in the main text:
> > In equation 5a (e.g. the smooth regularizer variant), this algorithm is an IJ estimate of the leave-one-out model with some additional noise. The amount of noise added to this approximation (determined by $c$) is dictated by the unlearning guarantees targeted by the company and properties of the function.  The variant designed for the unsmoothed version simply introduces the proximal function to the IJ estimate to handle the non-existence of the Hessian of the regularizer.

---

> > > ### Comment · Reviewer_EwhA · 2022-08-08
> > > **Thanks for the response.**
> > >
> > > I would like to thank the authors for justifications towards the application of their methods. Unfortunately, in its current form, I find the authors to overclaim or missing some key aspects of the paper. There seems to need significant changes to the paper.
> > > 1. The motivating examples of tech industry regulations do not seem to justify the study of linear models. Although the authors provided a reference about Airbnb models, it is rather old and hard to believe this is still being used as their production model. I do find the medical applications interesting and I would encourage the authors to elaborate on such motivation perspectives for future versions.
> > > 2. Although the algorithm is more efficient in the batched setting in the long run, no empirical results have been shown that the algorithm will outperform. There, I would expect a much smaller gap between the authors' algorithm and the baseline algorithm.
> > >
> > > I strongly encourage the authors to submit again in the future with the proper motivating examples and precise use cases.

---

> > > > ### Author Response · Authors · 2022-08-08
> > > > **misunderstanding the right to be forgotten**
> > > >
> > > > Can the reviewer elaborate more on what we are "overclaiming?" Our work is in keeping with a line of work making the same claims with similar motivation... . We are struggling to understand this reviewers remarks given the following.
> > > >
> > > > 1. As we have repeatedly clarified, **our paper does not study just linear models.** Our paper encompasses a large number convex optimization models unlike prior work. This means logistic regression, binary cross entropy, negative log likelihood, most generalized linear models etc. We make what models our algorithm applies to clear in the beginning of the paper
> > > >
> > > > Even if you find the Airbnb example to be old, we have provided several examples in healthcare and data pricing that are now included in the current draft. We feel there is no need for resubmission as our our work is consistent with a long line of work that study low-memory unlearning algorithms for convex ERM problems published at this same venue (and other top ML venues) with similar motivation. The problem of data removal is well motivated even in the convex setting! The reviewer seems to have the false impression that legislation and guidelines designed around the right to be forgotten should (or does) apply to only industrial sized models (which appears to allude to neural networks); but this is not the only applicable use case for unlearning algorithms. Can the reviewer provide citations that this is the case? The right to be forgotten concerns models that use user data and this includes several guidelines designed around datasets in healthcare, psychology, and information platforms that use convex models to train. We, and many philosophers, believe it should be applicable to most settings where user data is applied.
> > > >
> > > > We also want to mention that our algorithms outperform when trained on high-dimensional datasets (not large n) as the savings is in d which is not the canonical setting of neural networks. Please have a look at our updated draft as well as several of the other works in this space.
> > > >
> > > > 2. There also seems to be a fundamental misunderstanding regarding our experiments. As both examples are equivalent in both the batch and the online settings, the experiments shown are for the batch setting as well and therefore show the potential speedups after x months of data removal as well as the accuracy. Our experiments reflect settings that current data removal mechanism cannot be applied since they require smoothness. We have adjusted our captions to note this fact when it comes specifically to GDPR, but again, the right to be forgotten is not equivalent to GDPR and influences a lot more guidelines and policies.
> > > >
> > > > We urge the reviewer to reconsider their position as our work includes several key contributions and is well motivated as it is in keeping with a growing body of work in this area. At the very least, can the reviewer explain why motivation for just industrial sized models is necessary? We think this requirement from the reviewer is centered around their misunderstanding (or limited view) of the right to be forgotten and when it should be applied.

---

> > > > > ### Comment · Reviewer_EwhA · 2022-08-08
> > > > > **clarification**
> > > > >
> > > > > Can the authors provide a pointer to the captions you mentioned above? All of the legislations the authors mention in the introduction have a period of removal.
> > > > >
> > > > > I apologize for not noticing the changes in the new draft earlier. I am not saying it must include industrial scaled model. And I apologize for misspeaking earlier that the authors only study linear models. However, data removal on only convex models with online applications to medical domain seem limited compare to the entire space of applications needing data removal. And if the non-convex data deletion problem does not get studied in an online fashion, I don’t see these regulations shifting to require online deletion in the near future. That being said, I do find the authors’ algorithm useful in some real applications and will raise score once I see the GDPR comment properly included in the text/caption.

---

> > > > > > ### Author Response · Authors · 2022-08-08
> > > > > > **updates and further remarks**
> > > > > >
> > > > > > We appreciate the reviewers willingness to adjust their score and we have just updated figures 1, 2, and 3 and (lines 235-238) with our remarks about GDPR compliance.
> > > > > >
> > > > > > We still take slight issue with the characterization of our paper as performing “data removal on only convex models with online applications to medical domain.” We realize this impression as our sole contribution may have come from our paper’s title, where we have chosen to emphasize the online aspect of our algorithm. Regardless of acceptance, we plan to change the paper’s title to “Algorithms that approximate data removal: new results and limitations" so that all three contributions will be on equal footing. It is challenging to surface published convex ERM models being used in industrial applications because most of these are proprietary — however having worked and collaborated in these settings, we can attest that not all industrial applications which use user data and are under the purview of GDPR are non-convex ERM problems and our methods will be useful in these cases. To remind the reviewer of all of our contributions succinctly, they are:
> > > > > >
> > > > > > - An online/less expensive batch algorithm for data removal in convex ERM problems
> > > > > > - An algorithm for non-smooth convex ERM problems (which current approximate data removal algorithms cannot handle)
> > > > > > - A counter-example which implicates almost all established unlearning algorithms when hyper-parameter tuning has taken place
> > > > > >
> > > > > > We feel these contributions are more than enough to warrant acceptance at NeurIPS.
> > > > > >
> > > > > > Thanks!

---

> > > > > > > ### Comment · Reviewer_EwhA · 2022-08-08
> > > > > > > **Score update**
> > > > > > >
> > > > > > > I would like to thank the authors for improving upon the motivations of their work. I now find the utility of their algorithm justified. For the convex setting, I also find their contributions to be significant and sound. The adoption of infinitesimal jacknife and prox operator makes solid contribution upon previous work. The hyperparameter issue is the highlight of the paper as it introduces a new perspective towards this line of work.
> > > > > > >
> > > > > > > The scope of the problem (i.e. only convex loss functions) is the only reason why I’m not giving a 7. I would also like to encourage the authors to study non-convex settings in the future as that is potentially more impactful for the task of  data deletion.

---

### Official Review · Reviewer_p3VE · 2022-07-11

**Rating:** 5
**Confidence:** 4
**Soundness:** 3 good
**Presentation:** 2 fair
**Contribution:** 2 fair

**Summary:**

The paper proposes an online approximate unlearning algorithm that is efficient in computation and memory. To begin with, the paper uses the infinitesimal jackknife to enable the proposed algorithm to delete data from models trained which is non-smooth regularized. Furthermore, the paper theoretically proves the generalization and deletion capacity of the proposed method. Finally, extensive experimental results demonstrate that the proposed method can exceed in retraining the model and the second-order Taylor Approximation[1].
From my point of view, this paper falls into several significant drawbacks and might be improved by addressing the following issues:
1. It’s not clear how can the proposed method be effective in non-smooth regularized models.
2. The proposed method only inverts the Hessian for one time, but how can it ensure the total error is tolerable.
3. The supplementary lacks the comparison of runtime in different hyperparameter settings.
4. The candidates of the hyperparameter \lambda may be {10^{-3}, 10^{-4}, 10^{-5}, 10^{-6}} rather than {1^{-3}, 1^{-4}, 1^{-5}, 1^{-6}}.

[1] Ayush Sekhari, Jayadev Acharya, Gautam Kamath, and Ananda Theertha Suresh. Remember what you want to forget: Algorithms for machine unlearning. Advances in Neural Information Processing Systems, 34, 2021.


**Questions:**

1. How can the proposed method be effective in non-smooth regularized models?

**Limitations:**

The paper has not addressed the limitations of their work. It states that the proposed method fails to run with hyperparameter tuning, which is a common issue existing in unlearning algorithms.

**Strengths And Weaknesses:**

Strengths:
1. The paper provides sufficient theoretical proof of the proposed method.
2. According to the experimental results, the proposed method outperforms much better than the previously proposed unlearning algorithms.
Weaknesses:
1. The paper mainly enables the unlearning algorithm to be applied in non-smooth regularized models, but does not explain it specifically.
2. Lacking the comparison of runtime in different hyperparameter settings.

---

> ### Author Response · Authors · 2022-07-31
> **Thank you for reviews and response to review**
>
> We thank you for your reviews and the helpful comments provided. Below we provide responses to comments in the review:
>
> **How can the proposed method be effective in non-smooth regularized models?**
> Our proposed method is effective on non-smooth regularized models because we use the proximal operator, which is commonly used to optimize non-smooth ERM problems. We rely on software packages the glmnet and QUIC which provide efficient computation of the proximal operator for $\ell_1$ penalties to show our method is very efficient. Notably, the proximal operator effectively optimizes the regularizer within a small domain close to the current iterate, which can be done efficiently for simple penalties. This is what allows us to compensate for the lack of second derivative for the non-smooth penalty which hopefully explains why it works in non-smooth regularized models.
>
> Our algorithm does indeed work in this setting and we apologize for the extent to which the reasons why were not made clear. Our paper provides both theoretical guarantees  and experimental evidence that our method is effective for non-smooth regularized in Theorems 1 and 2 and Figure 2 respectively. Our work is the first to our knowledge to develop an algorithm which supports low memory unlearning of non-smooth regularized models. Even many of the memory intensive unlearning algorithms require smoothness [2]. Furthermore, our work is able to extend related work such as [1] to support non-smooth regularized models (Remark 1).
>
> **The proposed method only inverts the Hessian for one time, but how can it ensure the total error is tolerable:**
> Our error guarantee is statistically the same as other methods. This is due to the fact that the leave-one-out Hessian and full Hessian are sufficiently close (in particular $O(1/n)$ close). This, amount of closeness is enough to ensure that the model generated from our unlearning algorithm and the exactly unlearned model are $O(1/n^2)$ close. Adding noise on the order of $O(1/n^2)$ is what ensures the unlearning guarantee. The same kind of approximate unlearning argument is what is used  in previous works.  The statement of our unlearning guarantee for our IJ unlearning algorithm is in Theorem 1 and its proof is contained in Appendix A.3. We will provide additional text in the revision to make sure the structure of this argument is clear.
>
> **Comments about implementation**
> **Supplementary lacks the comparison of runtime in different hyperparameter settings:**
> We did not originally include the runtime metrics for the different hyperparameter settings as they did not affect the relative difference in runtimes. For completeness, we have included them in an updated version of the Appendix (please check our current revision) to confirm that the runtime remains the same across hyperparameters.
>
> **The candidates of the hyperparameter $\lambda$ may be $\{10^{-3}, 10^{-4}, 10^{-5}, 10^{-6}\}$ rather than $\{1^{-3}, 1^{-4}, 1^{-5}, 1^{-6}\}$:**
> Thank you for pointing out this mistake; we have now fixed it in the main text.
>
> [1] A. Sekhari, J. Acharya, G. Kamath, and A. T. Suresh. Remember what you want to forget: Algorithms for machine149
> unlearning. Advances in Neural Information Processing Systems, 34, 2021.
>
> [2] S. Neel, A. Roth, and S. Sharifi-Malvajerdi. Descent-to-delete: Gradient-based methods for machine unlearning. In145
> Algorithmic Learning Theory, pages 931–962. PMLR, 2021.

---

> > ### Author Response · Authors · 2022-08-07
> > **follow-up comment about stated limitations of our work**
> >
> > We address a few more of the reviewers specific points made about limitations of our work:
> >
> > - The reviewer writes that "the paper has not addressed the limitations of their work. It states that the proposed method fails to run with hyperparameter tuning, which is a common issue existing in unlearning algorithms."
> >
> > We would like to emphasize the three main limitations of our work; (1) our technique does provide guarantees of unlearning for most forms of non-convex optimization; (2) the guarantees do not extend to hyperparameter tuning (which is a limitation of all existing unlearning algorithms); and (3) our algorithm applies to non-smooth regularizers but not non-smooth objective functions. We have added to the paper making these limitations much clearer.
> >
> > In particular, for (1) we have highlight several potential use cases of our unlearning algorithm for industrial convex ERM problems (see comment for reviewer EwhA) but extending the influence function to work in nonconvex settings would have implications beyond privacy (as it is currently used for issues of explainability, robustness, and fairness) and would therefore be a significant step forward.  For (2) we have noted that our results about hyperparameter tuning is not just a limitation of our algorithm, but almost all unlearning algorithms that have been introduced so far. Again, this is an important limitation that we are actively pursuing, but also view pointing out this limitation as a contribution since it applies to more than just our algorithm. For (3) we view extensions of the influence function (aka infinitesimal jackknife) to non-smooth loss functions (and not just non-smooth regularizers) as an important one and are currently pursuing this as well.
> >
> > - The reviewer writes that "the paper mainly enables the unlearning algorithm to be applied in non-smooth regularized models, but does not explain it specifically."
> >
> > While we have added comments explaining how the non-smooth application through use of the proximal operator works, we note that our algorithm can be applied to smooth problems as well. In fact, our algorithm is more efficient than previously introduced methods for smooth problems after only two months of complying with GDPR (see response to reviewer EwhA).
> >
> > - The reviewer writes that the paper is "lacking the comparison of runtime in different hyperparameter settings."
> >
> > We hope we were able to address this concern with experiments we ran that are currently in the revision of our paper (see appendix E.1). The hyperparameter sweep makes no difference for the relative performance of the algorithms.

---

> > > ### Author Response · Authors · 2022-08-08
> > > **Have we addressed concerns?**
> > >
> > > Hi, we wanted to let you know that the paper has been updated and would like to know before the discussion period closes if we have addressed your concerns with our paper revisions and responses above.

---

> > > > ### Comment · Reviewer_p3VE · 2022-08-10
> > > > **Score Update**
> > > >
> > > > I would like to thank the authors' detailed response for addressing my concerns, but I have no time to check the correctness.
> > > >
> > > > Since all other reviewers have positive comments, I raise my score to 5.

---

> > > > > ### Author Response · Authors · 2022-08-10
> > > > > **remaining questions**
> > > > >
> > > > > We thank the reviewer for raising their score but we remain slightly confused.
> > > > >
> > > > > - The reviewer wanted us to add plots about the hyper-parameters and we have done so. What correctness is there to check? Shouldn't it be as easy as checking that we have done so?
> > > > >
> > > > > - If the reviewer hasn't checked correctness and are changing their score just because others have done so, should their confidence in their assessment be so high?
> > > > >
> > > > > - What outstanding concerns does the reviewer have? It seems the reviewer was confused about some key aspects of our work (our use of just one Hessian and how the prox operator bypasses the need for smoothness of the regularizer). Are those still concerns?
> > > > >
> > > > > We would really like to engage with this reviewer more to understand their assessments, but nevertheless thank them for changing their score to reflect slightly more positively on the paper.

---

### Official Review · Reviewer_K2o8 · 2022-07-13

**Rating:** 7
**Confidence:** 5
**Soundness:** 3 good
**Presentation:** 3 good
**Contribution:** 3 good

**Summary:**

The paper provides an algorithm for online data deletion. They consider unlearning with convex / strongly convex loss functions (with additional assumptions standard in the literature). The key improvement in the algorithm over prior works in [15, 29] is to provide an update step that does not require to recompute a new hessian inverse for every new deletion request (which they call infinitesimal Jacknife (IJ). Thus, they improve the running time from O(md^3) to O(md^2) where \(m\) is the number of deletions that can be handled. The authors provide extensions of this update step when the underlying regularization is not smooth (but the objective function is still smooth). This is obtained using a proximal update. Finally, the authors show an interesting limitation of unlearning for ERM based learning procedures.

**Questions:**

Refer to above comments

**Limitations:**

I do not see any negative societal consequences. In fact, the paper is trying to develop solutions for the important issues of right to be forgotten.

**Strengths And Weaknesses:**

Strength: The topic of machine unlearning is new and upcoming. The paper addresses computational side of the problem in the online deletion setting which is an important problem. The paper is well written!

Weaknesses:
1. I feel the key technical improvement from prior works is to come up with update in 5a in which H_l^{-1} does not need to be updated after every deletion request. While being interesting, I feel jacknife, proximal updates, etc are all well established theoretical tools and there is not much that fundamentally new, which is why I am not giving this paper a strong accept.


2.  I am a bit confused by the rationale to choose Definition 1 from [15] (instead of the smilar definition from [29]). In order for this definition to hold, one needs to randomize the out of the learning algorithm (which would decrease performance). In the corresponding definition in [29], one does not need the learning algorithm to be randomized. Furthermore, in your experiments where you report RT is this ERM + noise or just ERM ? (I feel there is a mismatch between the two sections).

3. Suppose we restrict outselves to generalized linear models ? In this case, the hessian is the sum of rank-1 matrices (one corresponding to each data point), and online updating the hessian as in [15, 29] is easy to do. What are the benefits of your algorithm in that case?

4. Do you have any suggessions from mitigating the reported issue in Section 5 for hyperparameter-tuned models ?

---

> ### Author Response · Authors · 2022-07-31
> **Thank you for reviews and response to review**
>
> We thank you for your reviews and helpful comments for our paper and appreciate the important impact of our results. Below we provide responses to concerns:
>
> **While being interesting, I feel jacknife, proximal updates, etc are all well established theoretical tools and there is not much that fundamentally new:**
> We appreciate that the reviewer found our application of the Jacknife to be interesting. While these tools are well established in the theory, our work is the first to demonstrate the applicability of the Jacknife to privacy concerns such as unlearning. More broadly, our work is in keeping with a line of research on different ways the IJ can be used for issues of societal concern. For example, *notable* (both papers won best paper awards) applications of the IJ (aka influence function) have focused on robustness [1] and explainability [2] concerns, and more recently, the jackknife was introduced as a way to ensure fairness [3].
>
> **Using the wrong definitions:**  In our experiments, we report ERM. We thank the reviewer for pointing out this mismatch. Our algorithm satisfies the definition of unlearning from [4] which we also prove in the Appendix. We have updated the paper that we focus on this definition so that our proofs and experiments are aligned.
>
> **Suppose we restrict ourselves to generalized linear models...What are the benefits of your algorithm in that case:** In this situation there will be little theoretical and practical difference between the Taylor approximation and the IJ approximation for unlearning datapoints. Our algorithm, however, will be beneficial for non-smooth GLM problems, which previous unlearning algorithms are not designed to handle. We demonstrate the theoretical benefits in Theorem 1 and the empirical runtime benefits in Figure 2.
>
> **Do you have any suggestions from mitigating the reported issue in Section 5 for hyperparameter-tuned models ?** We are currently working on this. We think it might be possible to augment the memory requirements of our unlearning algorithm or add more noise to the estimator to preserve the unlearning guarantees for specific types of hyperparameter tuning, but we are still in the beginning stages of proving such adjustments could work.
>
> [1] R. Giordano, W. Stephenson, R. Liu, M. Jordan, and T. Broderick. A swiss army infinitesimal jackknife. In The139
> 22nd International Conference on Artificial Intelligence and Statistics, pages 1139–1147. PMLR, 2019
>
> [2] P. W. Koh and P. Liang. Understanding black-box predictions via influence functions. In International conference143
> on machine learning, pages 1885–1894. PMLR, 2017.
>
> [3] E. Black and M. Fredrikson. Leave-one-out unfairness. In Proceedings of the 2021 ACM Conference on Fairness,135
> Accountability, and Transparency, pages 285–295, 2021.
>
> [4] A. Sekhari, J. Acharya, G. Kamath, and A. T. Suresh. Remember what you want to forget: Algorithms for machine149
> unlearning. Advances in Neural Information Processing Systems, 34, 2021

---

### Author Response · Authors · 2022-07-31
**Thank you for reviews!**

In this general comment, we would like to first thank the reviewers for taking the time to review the submission and the provided comments. Next, we would like to reiterate the contributions of this submission.

 Our work is in keeping with a line of research on different ways the infinitesimal jackknife (also called the influence function) can be used for issues of societal concern, including robustness, explainability, and fairness. To our knowledge, we are the first to introduce it in the context of privacy relating to data removal. For all prior works including ours, the IJ can be applied to non-convex settings but guarantees are virtually non-existent. Providing guarantees in a general non-convex setting would be a significant step forward for all these proposed applications but is outside the scope of our work. Our work notably provides guarantees for a broad range of models generated from both smooth and non-smooth convex ERM problems (e.g. logistic regression, hinge loss, cross entropy loss etc.) as well as specialized non-convex models (see line 144) as was done in previous works. In light of the feedback given, we elaborate on our three contributions:

* **An online algorithm:** while the current form of GDPR requires deletion after a month to allow companies time to comply, showing it can be done efficiently and quickly could incentivize a more narrow window for compliance. This, in turn, could mitigate the harm done from potential privacy attacks even further than current legislation. Notably, even in the batch setting our algorithm will be more efficient than all low-memory unlearning algorithms in the literature so far after only two months of complying with requests since it requires inverting a Hessian once and storing it while other methods will require inverting a Hessian monthly. This, we believe, would  more than likely make the IJ approximation the standard technique used for compliance with data removal requests (at least when compared with the presently introduced techniques).
* **Application to non-smooth models:** we provide theoretical guarantees of data deletion and experimental result for models with non-smooth regularizers such as the $\ell_1$ penalty. To the best of our knowledge, no current approximate unlearning algorithm and very few memory intensive unlearning algorithms can be applied in this popular setting. Our technique for handling non-smoothness can be applied to the previously introduced Taylor approximation (TA) and we provide theoretical guarantees for this.
* **Counterexample:** Hyper-parameter tuning is common in most model generation pipelines. Our work is the first to point out that unlearning via all techniques presented so far does not work when this procedure takes place. While this is the last part of the paper, we feel this contribution alone is a noteworthy one for the community as it could likely prevent unintended harm.
\end{itemize}

We will make all these contributions clearer in the revision. Finally, there were additional concerns regarding motivation of our algorithm, clarity about why the proximal IJ works, and experimental details. We give more detailed responses to each reviewers concerns below.

---

### Public Comment · ~Eli_Chien1 · 2023-09-19
**Publicly available code?**

Dear authors,

Thank you for the great work, I am very interested in it. I would like to ask if there is any way that I can get the code to reproduce the experiment? The authors mentioned in the paper that the code is attached separately but the Supplementary Material on OpenReview is just a pdf. I also did not find any Github link in the paper by searching the keyword github. Maybe I overlooked it, but it would be a great help if the authors could provide their code.

Thanks,

Eli

---

> ### Public Comment · ~Vinith_Menon_Suriyakumar1 · 2023-10-03
> **Code**
>
> Hi Eli,
>
> Thanks for your interest! Apologies on our end, it’ll be on GitHub by Thursday morning. I’ll respond with the link here!
>
> Best,
> Vinith

---

> > ### Public Comment · Authors · 2023-10-05
> > **Code available**
> >
> > Here is the link: https://github.com/VMS-6511/online-data-deletion. Let us know if you have any issues!

---

### Meta-Review · Area_Chair_nM4g · 2022-08-27

**Recommendation:** Accept
**Confidence:** Less certain

**Metareview:**

Most of the reviewers agree that this paper is well written and provides a notable improvement over prior works on algorithms for data deletion. Some initial concerns regarding the proper motivation for the problem setting have been largely addressed.

**Award:**

No

---

### Decision · Program_Chairs · 2022-09-14

Accept